# Light-wave-controlled Haldane model in monolayer hexagonal boron nitride

Sambit Mitra[1,2], Álvaro Jiménez-Galán[3,4 ✉], Mario Aulich[2,5], Marcel Neuhaus[1,2,5], Rui E. F. Silva[3], Volodymyr Pervak[1,2], Matthias F. Kling[1,2,5,6] & Shubhadeep Biswas[1,2,5 ✉]

In recent years, the stacking and twisting of atom-thin structures with matching crystal symmetry has provided a unique way to create new superlattice structures in which new properties emerge[1,2]. In parallel, control over the temporal characteristics of strong light fields has allowed researchers to manipulate coherent electron transport in such atom-thin structures on sublaser-cycle timescales[3,4]. Here we demonstrate a tailored light-wave-driven analogue to twisted layer stacking. Tailoring the spatial symmetry of the light waveform to that of the lattice of a hexagonal boron nitride monolayer and then twisting this waveform result in optical control of time-reversal symmetry breaking[5] and the realization of the topological Haldane model[6] in a laser-dressed two-dimensional insulating crystal. Further, the parameters of the effective Haldane-type Hamiltonian can be controlled by rotating the light waveform, thus enabling ultrafast switching between band structure configurations and allowing unprecedented control over the magnitude, location and curvature of the bandgap. This results in an asymmetric population between complementary quantum valleys that leads to a measurable valley Hall current[7], which can be detected by optical harmonic polarimetry. The universality and robustness of our scheme paves the way to valley-selective bandgap engineering on the fly and unlocks the possibility of creating few-femtosecond switches with quantum degrees of freedom.

Discrete symmetries in quantum mechanics, namely spatial inversion and time reversal, play a pivotal role in governing the electronic and topological properties of materials[6,8,9]. For example, when both the symmetries are preserved in hexagonal monolayers such as graphene, the highest valence and lowest conduction bands are connected by Dirac-like dispersion at the high-symmetry K and K′ points of the Brillouin zone, enabling semimetal-like behaviour[9]. When the spatial-inversion symmetry is broken, foar example, in monolayer hexagonal boron nitride (hBN) (Fig. 1a), a bandgap ($\Delta$) appears along with a parabolic band dispersion around the K and K′ points, leading to insulating behaviour ($\Delta_{K,K'} \approx 5.9$ eV for hBN)[10]. The two energy-degenerate local band minima at the K and K′ points are quantum valleys and are time-reversal partners. When time-reversal symmetry is broken, carrier dynamics at K and K′ can be distinguished (Fig. 1b). One paradigmatic example is the toy model of a Chern insulator proposed by F. D. M. Haldane[6]. Here time-reversal symmetry is envisioned as being broken by a staggered magnetic field in gapped graphene, thus inducing complex next-nearest neighbour (CNNN) hoppings that modify the relative bandgap at the valleys (Fig. 1c). Another relevant example is the work of Xiao et al.[11], who proposed valley-selective excitation by resonant circularly polarized light in gapped graphene, which has been demonstrated in various two-dimensional transition metal dichalcogenides[12,13], developing the field of valleytronics[12–19].

Controlling the spatial-inversion and time-reversal symmetries allows us to engineer the material properties, including those not found naturally in pristine materials. By vertical stacking of two-dimensional materials with matching spatial symmetry, we can create composite multilayer structures with a different symmetry and electronic properties than those of the individual layers[20], which can be further manipulated by controlling the stacking angle[1,2]. Moreover, applying external time-dependent electric and magnetic fields allows us to induce and control transient material properties that may not be naturally found in thermal equilibrium. Previous works have shown using Floquet theory that the frequency, intensity and helicity of long-pulse-duration electric fields can be exploited as additional degrees of freedom to modify the Hamiltonian parameters of a crystal during an interaction[21–24] on the timescale of the pulse duration[25]. In light-wave electronics[3,4,26–40], one exploits control over the temporal characteristics of strong light fields, such as the carrier envelope phase of single-cycle pulses or the time delay between multicolour fields, to manipulate coherent electronic transport on timescales shorter than one laser cycle.

Here we introduce a light-wave-driven counterpart to twisted layer stacking in hexagonal materials. We use the matching between the threefold symmetry of the crystal lattice and the structured spatial waveform of the light field, a degree of freedom that we control with sublaser-cycle precision (Fig. 1a). By rotating the structured light wave in space, we demonstrate subcycle-controlled time-resolved

[1]Max Planck Institute of Quantum Optics, Garching, Germany. [2]Physics Department, Ludwig-Maximilian University of Munich, Garching, Germany. [3]Instituto de Ciencia de Materiales de Madrid (ICMM), Consejo Superior de Investigaciones Científicas (CSIC), Madrid, Spain. [4]Max Born Institute, Berlin, Germany. [5]SLAC National Accelerator Laboratory, Menlo Park, CA, USA. [6]Department of Applied Physics, Stanford University, Stanford, CA, USA. ✉e-mail: alvaro.jimenez@csic.es; shubha@slac.stanford.edu

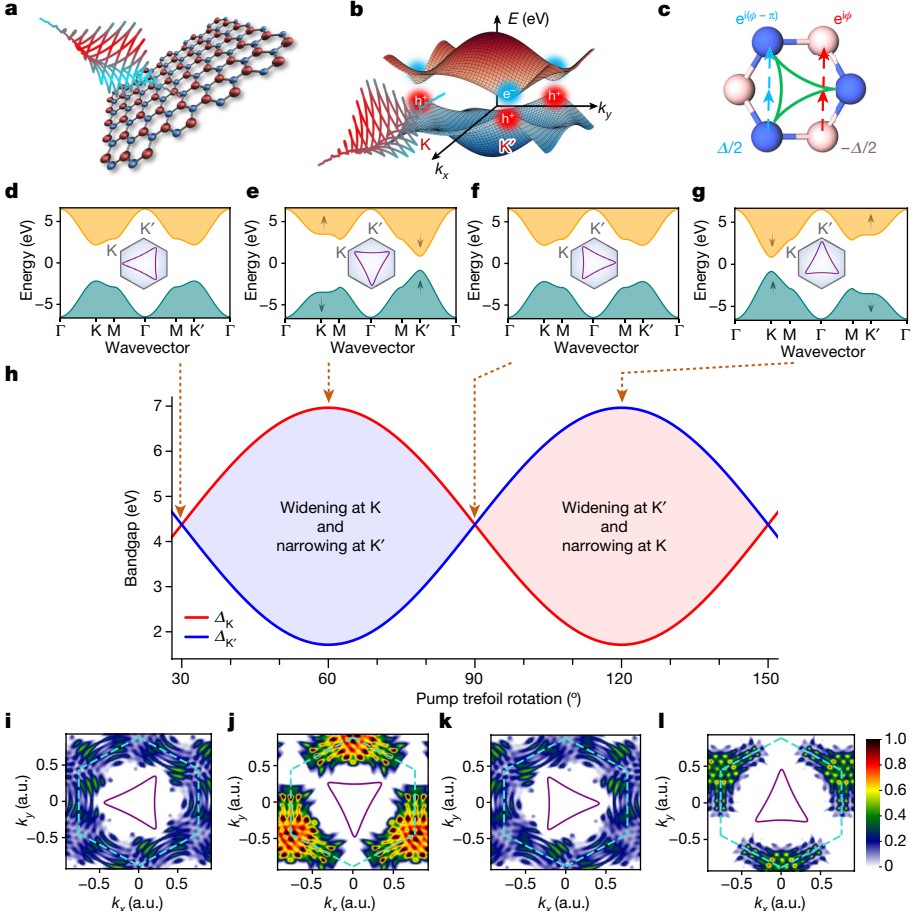

**Fig. 1 | Light-wave-controlled valley-selective bandgap modification.**
**a**, The tailored intense light waveform, which resembles a trefoil structure on the lattice plane, is used to coherently manipulate the band structure of monolayer hBN. **b**, For such a two-dimensional system, the band structure shows hexagonal symmetry with a minimum bandgap at the K and K′ points (valleys) at the vertices of the hexagon. In monolayer hBN, the optical bandgap is $\Delta \approx 5.9$ eV. **c**, Upon interaction with a strong trefoil waveform, the couplings between the atoms are modified. The laser-dressed system is described by a Haldane-type tight-binding Hamiltonian, such that the next-nearest neighbour hoppings are complex and carry a phase difference of π between the two sublattices, with on-site energies $\Delta/2$ and $-\Delta/2$. **d–g**, As the field (and its associated vector potential shown with respect to the first Brillouin zone) rotates in space, the CNNN hoppings are modified, changing from purely real

(**d**,**f**) to purely imaginary (**e**,**g**). In **d** and **f**, the waveform points in between K and K′ (that is, M), but with different orientations. In **e** and **g**, it points to K and K′, respectively. The latter lifts the degeneracy between the K and K′ valleys, reducing the bandgap in one of them and increasing it in the other. **h**, The effective bandgap oscillates as a function of the rotation of the trefoil light field, leading to different electron excitation dynamics between the valleys. **i–l**, Normalized electron populations in the conduction band corresponding to the trefoil field orientation shown in the inset of **d–g**. This is calculated using the code described in ref. 44. The green hexagons delimit the first Brillouin zones, with the K and K′ points at the vertices, as indicated in the inset of **d–g**. The purple triangles indicate the orientation of the vector potential. The electron population is a maximum in the valley where the bandgap is lower during the laser–matter interaction. a.u., arbitrary units.

symmetry breaking and valley bandgap control in an insulating hBN monolayer, thus realizing a light-driven analogue to the topological model of Haldane. Our scheme enables symmetry-protected and selective bandgap manipulation on the fly for light-wave electronics and two-dimensional materials engineering, which we exploit here to demonstrate a below-resonant, strong-field regime for ultrafast valleytronics in an insulating quantum system.

## Light analogue of the Haldane model

Two counter-rotating circularly polarized fields of frequencies $\omega$ and $2\omega$ combine to produce a strong tailored light wave whose projection on the hBN crystal plane resembles a trefoil structure, which matches the triangular lattice of hBN. Such a field induces CNNN hoppings in the hBN monolayer during the laser–matter interaction[5] (Fig. 1c and Methods). In analogy to the topological model of Haldane[6], the laser-induced CNNN hoppings break time-reversal symmetry and lift the degeneracy of the valleys, thus reducing the bandgap in one and increasing it in the

other depending on the CNNN hopping angle $\phi$ (Fig. 1c). The latter is controlled by the orientation of the vector potential of the waveform (Methods). Changing the phase delay between the $\omega$ and $2\omega$ pulses rotates the electric field and the vector potential and modifies the CNNN hopping and band structure, as shown in Fig. 1d–g (ref. 5 and Methods). Owing to the periodicity of both the trefoil waveform and the lattice, the same dynamics are repeated every 120°. Figure 1h shows the predicted variation of the effective bandgap at the K and K′ valleys for different orientations of the vector potential relative to the Brillouin zone of the hBN crystal, which exceeds 5 eV for our experimental conditions (laser field strength $F_L \approx 0.7$ V Å⁻¹ and frequency $\omega = 0.6$ eV).

The light field that modifies the band structure is strong and has a frequency well below the equilibrium bandgap ($\omega \ll \Delta_{K,K'} = 5.9$ eV), which drives tunnel excitation of carriers from the valence to the conduction band close to the effective minimum bandgap. Thus, in the presence of the laser-induced valley-contrasting band structure modification indicated above, the resulting valley populations should oscillate as the trefoil waveform is rotated, with strong K′ (K) valley

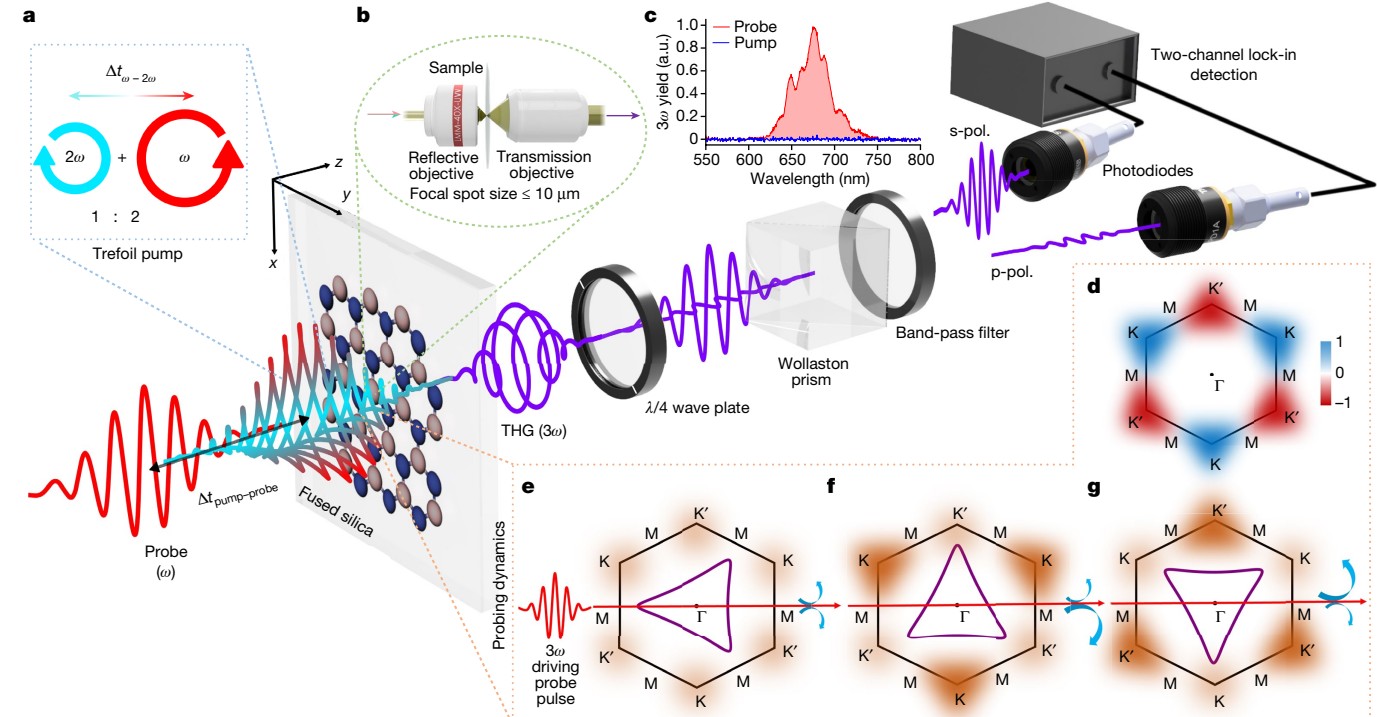

**Fig. 2 | All-optical methodology to control and probe the bandgap engineering with application to valleytronics. a**, Monolayer hBN mounted on a 500 μm, thin, fused-silica substrate was pumped with a 30-fs-long trefoil waveform. The light wave was generated by interferometrically combining a counter-rotating circularly polarized 2 μm ($\omega$) and 1 μm ($2\omega$) wavelength light with an intensity ratio of 2:1. Its different orientation with respect to the real-space lattice orientation was achieved by controlling the subcycle delay ($\Delta t_{\omega-2\omega}$) between the $\omega$ and $2\omega$ pulses. A time-delayed ($\Delta t_{pump-probe}$), linearly polarized, 2 μm wavelength pulse measures the pump-induced valley dynamics through time-resolved harmonic polarimetry of its third harmonic. The polarization state of the generated third harmonic, analysed by a quarter-wave plate and Wollaston prism, encode the information about the induced valley polarization. Eventually, the spectrally filtered and specially separated s-polarized (s-pol.) and p-polarized (p-pol.) components are captured with photodiodes, which are connected to a two-channel lock-in amplifier for data acquisition. **b**, The microscopic geometry of the interaction restricts it within a region of 10 μm, comparable to that of the monocrystalline patch. **c**, $3\omega$ yield as a function of wavelength for pump (blue) and probe (red) irradiation. **d**, Sketch of Berry curvature in the conduction band of monolayer hBN. **e–g**, Three representative probing configurations. **e**, The K and K' valleys are equally populated, resulting in a net-zero anomalous Hall current (depicted with curved blue arrows) and a linearly polarized third harmonic. **f,g**, The asymmetry in the electron populations between the K and K' valleys results in a non-zero Hall current, creating an elliptical third-harmonic signal with valley-dependent helicity. In **f**, the K valley is more populated, whereas the K' valley is more populated in **g**, resulting in dominance of opposite helicities (indicated by asymmetric curved arrows) in the third harmonic.

polarization when the vector potential is oriented as outlined in Fig. 1e (Fig. 1g). We performed time-dependent simulations for a two-band model of gapped graphene using the crystal parameters of monolayer hBN under our experimental conditions. Figure 1i–l shows the electron populations calculated after the interaction with a trefoil field with fixed helicity for the four spatial orientations outlined in Fig. 1d–g. Electrons predominantly populate the valley where the bandgap is reduced, highlighting valley-contrasting band modification.

Figure 1i–l shows that two visible simultaneous mechanisms play a role in the overall valley polarization. On the one hand, the bandgaps at the K and K' valleys become asymmetric owing to the tailored strong light field interaction, as discussed above. This effect is dependent only on the orientation of the vector potential relative to the lattice. On the other, the helicity of the driving field (and its associated vector potential) induces valley-contrasting dynamics, as in conventional valleytronics[11–13], and is independent of the rotation angle. In our regime, the former largely dominates the valley excitation dynamics, as can be seen by the valley population oscillating and switching as a function of the trefoil orientation for a fixed helicity. There are, however, visible effects of the helicity-dependent valley circular dichroism apparent in the simulations. In particular, the electron populations at K and K' do not merely flip between $\theta = 60°$ and $\theta = 120°$ (compare with Fig. 1j,l). For a complete flip between K and K', one must change both

the orientation of the vector potential and its helicity, which provides the time-reversal partner field. Still, valley polarization can be clearly switched without changing the field helicity by rotating the waveform, as is evident in Fig. 1j,l, and its oscillation provides a telltale sign of the modification of the band structure topology.

## Subcycle-controlled, non-resonant valleytronics

In the experiment, we prepared the pump trefoil waveform by interferometrically combining two counter-rotating, circularly polarized, light waves of about 30 fs and with an intensity ratio of 2:1 (Fig. 2a). The broadband spectra (Extended Data Fig. 7) of these mutually phase-stable components were centred around 2 μm (frequency $\omega$ and photon energy 0.6 eV) and 1 μm (frequency $2\omega$ and photon energy 1.2 eV), respectively. We used a pump intensity of about 4–7 TW cm$^{-2}$ in air without observing any damage. The orientation of the trefoil waveform with respect to the fixed hBN lattice structure was controlled by the subcycle time delay between the $\omega$ and $2\omega$ light waves. Further details are given in Methods.

To measure the valley polarization as a function of the pump trefoil rotation, we measured the valley-dependent changes induced by the pump on the polarization of the nonlinear (third harmonic) response of a time-delayed, linearly polarized, probe light field. All-optical probing

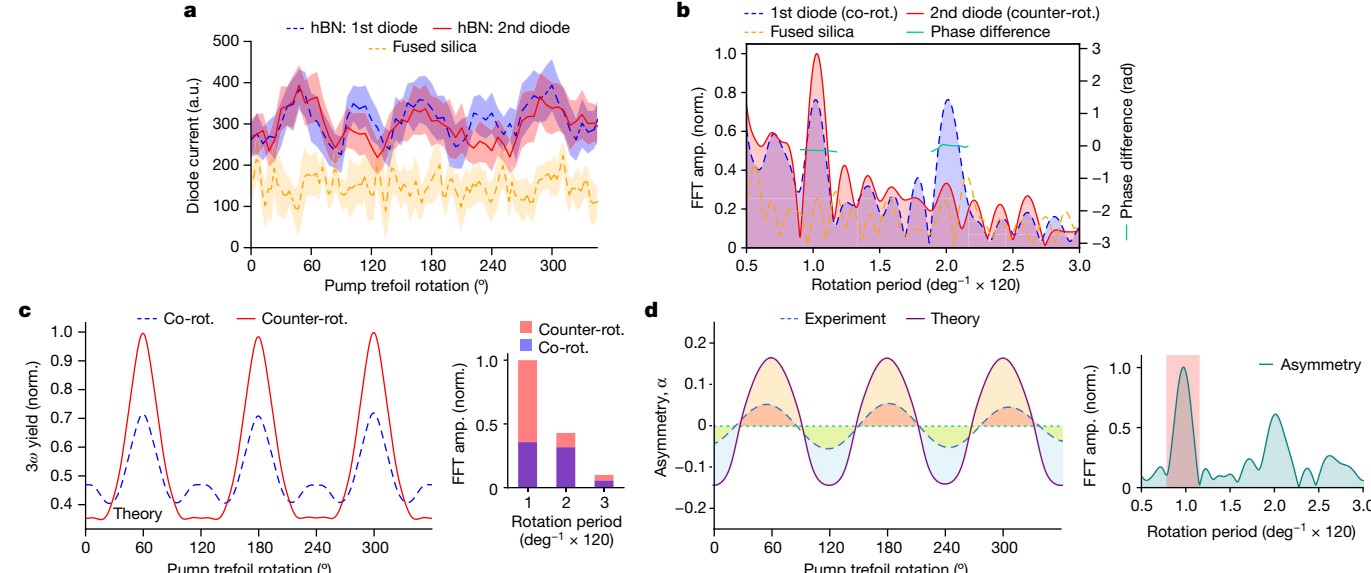

**Fig. 3 | Comparison between experimental and theoretical results.**
**a**, Experimentally obtained helicity-resolved third-harmonic (3ω) yield as a function of pump trefoil rotation. The blue and red curves present data of two photodiode currents measured during the exposure of the hBN sample. Representative data for exposure of only the fused-silica substrate is represented by the yellow curve. The solid line was produced by taking the statistical mean of all the photodiode current data points recorded within a bin size of 3°. The shaded regions depict the respective standard deviations. **b**, Fast Fourier transform (FFT) spectra of the curves in **a**. Distinct peaks corresponding to a 120° periodic oscillation are present for hBN (blue and red) and absent for the substrate (yellow). The phase difference between the oscillations from the two photodiodes for hBN is represented by the green curves. **c**, Simulated 3ω yields corresponding to a helicity co-rotating (blue curve) and counter-rotating (red curve) with the ω field of the trefoil as a function of the pump trefoil rotation. The inset shows the FFT spectra of the curves. Calculations were performed for a pump intensity of 7 TW cm⁻². **d**, Comparison of the asymmetries (α) deduced from the experimental and theoretical data. The experimental curve was obtained after Fourier filtering around the frequency corresponding to the 120° oscillation, as shown in the inset. amp., amplitude; norm., normalized; rot., rotating.

offers simplicity and potential for seamless integration into future all-optical photonic technologies. However, note that other techniques, such as angle-resolved photoemission spectroscopy, X-ray absorption spectroscopy or measurements of the transport valley Hall current, can be also applied to probe the valley population in the present scheme.

We separated a small portion of the fundamental 2 μm (ω) beam, with linear polarization and a pulse duration of about 30 fs, and delayed it by approximately 60–120 fs with respect to the trefoil pump pulse. The linearly polarized probe field $E_{\text{probe}}$ generated a drift current along the polarization axis, as well as an anomalous (orthogonal) current that was proportional to the population asymmetry at the complementary valleys[5,7,15,41]:

$$j_{\text{anomalous}}(t) = \frac{e^2}{\hbar(2\pi)^2} E_{\text{probe}}(t) \sum_n \int_{\text{BZ}} \mathrm{d}\mathbf{k}\, \rho_{nn}(\mathbf{k}, t) \Omega_n(\mathbf{k}),$$

where $e$ is the electron charge, $\rho_{nn}$ is the population of band $n$ and $\Omega(\mathbf{k})$ is its Berry curvature.

The drift current is governed by the vector potential of the field, whereas the anomalous current is proportional to the electric field. This leads to a π/2 phase-delay difference between these components and, thus, to the emission of elliptically polarized harmonic radiation. Note that the Berry curvatures around the K and K′ valleys exhibit opposite signs (Fig. 2d), $\Omega_n(\text{K}) = -\Omega_n(\text{K}')$. In the absence of valley polarization, $\rho_{nn}(\text{K}) = \rho_{nn}(\text{K}')$, the anomalous current cancels, and the nonlinear optical response induced by the probe field is linearly polarized. In contrast, when there is valley polarization, $\rho_{nn}(\text{K}) \neq \rho_{nn}(\text{K}')$, the anomalous current is non-zero and its sign is opposite for K and K′ valley polarization. This makes the optical response of the probe elliptical and with opposing helicity for K and K′ valley polarization (Fig. 2e–g). Hence, the valley polarization induced by the trefoil pump field is mapped onto the asymmetry $\alpha = \frac{S_1 - S_2}{S_1 + S_2}$ of the two helicity components $S_1$ and $S_2$ of the probe harmonic (3ω in the present case)[5,42]. Experimentally, we determined

both the helicity and ellipticity by measuring the s- and p-polarized components of the outgoing 3ω probe radiation by two photodiodes following a combination of a quarter-wave plate and a Wollaston prism. During the experiment we rotated the pump trefoil waveform for a fixed delay and polarization of the probe pulse; the sample and other parameters of the experiment were kept fixed.

When the laser irradiated only the fused silica, we observed no characteristic modulation as a function of the pump trefoil rotation and for a fixed pump–probe delay, as shown in Fig. 3a (orange curve). In striking contrast, when the hBN sample was irradiated, both diodes displayed clear oscillating signals with a dominant 120° periodicity (red and green curves in Fig. 3a). The Fourier analysis in Fig. 3b confirms strong 120° periodic oscillations, with one of the diodes also showing a contribution of its second order (60° periodic oscillation). All these experimental features are fully consistent with the interpretation given above and reproduced by time-dependent simulations on a gapped-graphene system, as shown in Fig. 3c. As in the experiment, the simulations measure the 3ω helicity signal of a probe field that is co-rotating (green curve) or counter-rotating (red curve) with the ω field of the trefoil, which correspond to the signals $S_1$ (green curve) and $S_2$ (red curve), respectively, measured at the diodes.

First, we discuss the origin of the 60° oscillation. Comparing Fig. 1j and Fig. 1l, we see that the population at the K valley oscillates less than that at K′. As we mentioned earlier, this reflects that a 60° rotation of the field alone does not produce the time-reversal partner field; for this, one must change the helicity of the field as well. Thus, the valley polarization oscillation as a function of pump rotation is not a perfect sinusoidal curve of 120° periodicity and leads to the appearance of higher-order Fourier components (see Methods and Extended Data Fig. 4 for further details).

We now focus on the 120° oscillation by Fourier filtering this component only. In particular, we are interested in the asymmetry parameter $\alpha$ of the two helicities, which maps the valley polarization. This is

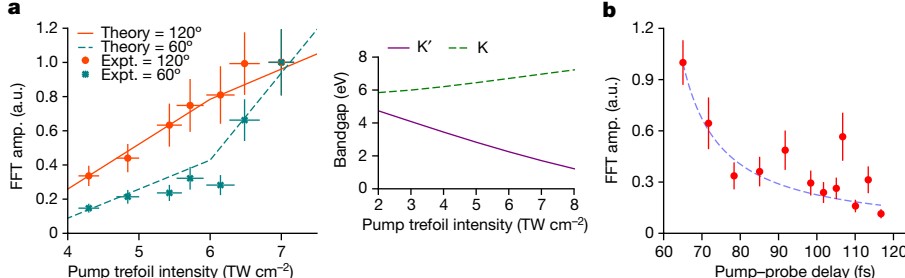

**Fig. 4 | Pump intensity scaling of bandgap modification and temporal dynamics of valley polarization. a**, Fourier amplitudes of the $3\omega$ co-rotating signal as a function of pump intensity. Red corresponds to the signal at 120° period, green to that at 60° period. Lines present the theoretical simulations. Inset, Pump intensity scaling of the bandgap at the K (purple, solid) and K′ (green, dashed) valley for a trefoil orientation along $\theta = 120°$. **b**, Fourier amplitude of the $3\omega$ co-rotating signal at 120° period as a function of the pump–probe delay. The dashed line serves as guide for the eye. The error bars represent standard deviations for an ensemble of measurements. Expt., experimental.

shown in Fig. 3d. This parameter changes sign in both the experimental and theoretical data, which signals a switch in the valley polarization. The asymmetry curve oscillates from a positive maximum at $\theta = 60°$, corresponding to K′ valley polarization (Fig. 1j), to a negative maximum at $\theta = 120°$, corresponding to K valley polarization (Fig. 1l), and is zero in the absence of valley polarization, for $\theta = 30°$ and $90°$. The oscillation and switch of the valley polarization as a function of the pump trefoil rotation, for a fixed pump helicity, are evidence of the trefoil-field-induced modification of the band structure.

## Light-induced valley bandgap reduction

Figure 4a shows the scaling of the oscillation amplitude of the $3\omega$ signal of the linearly polarized probe with the intensity of the pump trefoil field. As the pump trefoil intensity is increased, the theory predicts a larger bandgap asymmetry at the valleys (inset in Fig. 4a) and, thus, a larger valley polarization. This observation is reproduced in the experiment. Furthermore, the strength of the 60° oscillations increases with increasing trefoil field strength, reaching the same value as the 120° oscillation at around 7 TW cm$^{-2}$. This suggests that there is a gradual increase of the effect of helicity-dependent valley physics, which should be stronger when the effective bandgap approaches a value comparable to the photon energy.

## Strong-field-induced valley lifetime

We additionally performed experiments at various pump–probe time delays. Figure 4b shows the Fourier amplitude of the 120° oscillation of the $3\omega$ probe signal for pump–probe delays between 60–120 fs. For delay times shorter than 60 fs, we obtained a random signal in the Fourier analysis due to the temporal overlap between the pump and probe fields, which distorts the trefoil structure. Above 60 fs, the signal gradually decreases until 120 fs, where it falls below the noise floor. Thus, we estimate that valley population asymmetry in hBN persists for about 60 fs following a strong-field excitation. Note that this timescale might differ from that measured after one-photon resonant excitations, for which the electron distribution is more localized in the valleys[16]. All measurements were performed at ambient temperature.

## Conclusion and outlook

In summary, we have demonstrated that a below-resonant strong light field with polarization tailored to the crystal symmetry induces a waveform-controlled CNNN hopping that breaks time-reversal symmetry in a laser-dressed hBN monolayer. In this way, we realized the light analogue of the Haldane model with the parameters of this model controlled on sub-laser-cycle timescales. We showed that the rotation of the waveform modifies the magnitude and location of the bandgap, enabling non-resonant valley polarization switching in an insulating monolayer of hBN that we measured using time-resolved optical harmonic polarimetry. These experiments extend valleytronics to the few-femtosecond timescale and to a wider range of materials, which may include combinations of semiconducting and insulating monolayers, that have been out of reach due to large optical bandgaps or symmetry constraints. Indeed, in parallel to this work, Tyulnev et al.[43] demonstrated strong-tailored-field-induced valley manipulation in bulk MoS$_2$. Additionally, we estimated that hBN has a valley lifetime of about 60 fs at ambient temperature when driven by strong, tailored light fields. This is around nine cycles of our optical field. These results may encourage future studies of room-temperature ultrafast switches with strong tailored light. The waveform-driven, valley-contrasting, bandgap modification is symmetry-robust, subfemtosecond-controllable and reversible, and it preserves electron coherence, thus opening a new route towards the ultrafast control of material properties. Our work may offer a non-equilibrium, light-driven counterpart to twisted layer stacking, for example by patterning the light wave such that the trefoil shape rotates spatially along the beam profile, thus creating micrometre-sized domains in the material with different electronic properties, like Moiré patterning.

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

## Methods

### Theoretical model and simulations

The model was described in detail in ref. 5. Here we provide further details of its derivation and application to our case. Atomic units are used unless otherwise stated.

### Definition of the lattice.

Hexagonal boron nitride is formed from two triangular sublattices A and B, which host boron and nitrogen atoms, respectively. Extended Data Fig. 1a shows the lattice up to next-nearest neighbour atoms. The two lattice vectors can be defined as

$$\mathbf{a}_1 = \frac{r_0}{2}(3, \sqrt{3})$$
$$\text{and} \quad \mathbf{a}_2 = \frac{r_0}{2}(3, -\sqrt{3}), \tag{1}$$

where $r_0$ is the distance between nearest neighbours. The distance from atom $j$ to atom $i$ can be written in terms of $r_0$ and the angle $\alpha$ between those two atoms (Extended Data Fig. 1a):

$$[r_{ij,x}, r_{ij,y}] = (\sqrt{3})^m r_0 [\sin\alpha_{ij}, \cos\alpha_{ij},], \tag{2}$$

where $m = 0, 1$ for nearest neighbours and next-nearest neighbours, respectively. The Brillouin zone of hBN is shown in Extended Data Fig. 1b, along with its high-symmetry points.

### Definition of the field.

We start with a bicircular field vector, which results from the combination of two counter-rotating circular fields of frequencies $\omega$ and $2\omega$. We define it as

$$\mathbf{F}_{\circlearrowright}(t) = [F_x(t), F_y(t)] = [F_\omega\sin(\omega t) - F_{2\omega}\sin(2\omega t + \varphi),$$
$$F_\omega\cos(\omega t) + F_{2\omega}\cos(2\omega t + \varphi)]. \tag{3}$$

The field strengths of the fundamental and second-harmonic fields are $F_\omega$ and $F_{2\omega}$, respectively, and $\varphi$ is the phase delay between the two fields. From the electric field, we define the vector potential $\mathbf{A}(t) = -\int dt\, \mathbf{F}(t)$, so that

$$\mathbf{A}_{\circlearrowright}(t) = [A_x(t), A_y(t)] = \left[\frac{F_\omega}{\omega}\cos(\omega t) - \frac{F_{2\omega}}{2\omega}\cos(2\omega t + \varphi),\right.$$
$$\left. -\frac{F_\omega}{\omega}\sin(\omega t) - \frac{F_{2\omega}}{2\omega}\sin(2\omega t + \varphi) \right]. \tag{4}$$

With this definition, both the electric field and the vector potential rotate clockwise. When we switch the helicities of the two circular fields such that $[F_x(t), F_y(t)] \rightarrow [-F_x(t), F_y(t)]$, we obtain

$$\mathbf{F}_{\circlearrowleft}(t) = [-F_\omega\sin(\omega t) + F_{2\omega}\sin(2\omega t + \varphi),$$
$$F_\omega\cos(\omega t) + F_{2\omega}\cos(2\omega t + \varphi)] \tag{5}$$

and

$$\mathbf{A}_{\circlearrowleft}(t) = \left[-\frac{F_\omega}{\omega}\cos(\omega t) + \frac{F_{2\omega}}{2\omega}\cos(2\omega t + \varphi),\right.$$
$$\left. -\frac{F_1}{\omega}\sin(\omega t) - \frac{F_2}{2\omega}\sin(2\omega t + \varphi) \right]. \tag{6}$$

In this case, both the electric field and vector potential rotate anticlockwise. Between $\mathbf{F}_{\circlearrowright}$ and $\mathbf{F}_{\circlearrowleft}$, the field has rotated in space by 180° (compare the green curves in Extended Data Fig. 2a,d). However, the orientation of the vector potential is the same for $\mathbf{A}_{\circlearrowright}$ and $\mathbf{A}_{\circlearrowleft}$ (compare the purple curves in Extended Data Fig. 2b,e).

For generality, let us also consider the bicircular field defined as

$$\widetilde{\mathbf{F}}_{\circlearrowright}(t) = [\widetilde{F}_x(t), \widetilde{F}_y(t)] = [F_\omega\cos(\omega t) + F_{2\omega}\cos(2\omega t + \varphi),$$
$$-F_\omega\sin(\omega t) + F_{2\omega}\sin(2\omega t + \varphi)] \tag{7}$$

and, consequently,

$$\widetilde{\mathbf{A}}_{\circlearrowright}(t) = [\widetilde{A}_x(t), \widetilde{A}_y(t)] = \left[-\frac{F_\omega}{\omega}\sin(\omega t) - \frac{F_{2\omega}}{2\omega}\sin(2\omega t + \varphi),\right.$$
$$\left. -\frac{F_\omega}{\omega}\cos(\omega t) + \frac{F_{2\omega}}{2\omega}\cos(2\omega t + \varphi) \right]. \tag{8}$$

When we perform the following switch of the circular field helicities $[\widetilde{F}_x(t), \widetilde{F}_y(t)] \rightarrow [\widetilde{F}_x(t), -\widetilde{F}_y(t)]$,

$$\widetilde{\mathbf{F}}_{\circlearrowleft}(t) = [F_\omega\cos(\omega t) + F_{2\omega}\cos(2\omega t + \varphi),$$
$$F_\omega\sin(\omega t) - F_{2\omega}\sin(2\omega t + \varphi)] \tag{9}$$

and

$$\widetilde{\mathbf{A}}_{\circlearrowleft}(t) = \left[-\frac{F_\omega}{\omega}\sin(\omega t) - \frac{F_{2\omega}}{2\omega}\sin(2\omega t + \varphi),\right.$$
$$\left. \frac{F_\omega}{\omega}\cos(\omega t) - \frac{F_{2\omega}}{2\omega}\cos(2\omega t + \varphi) \right]. \tag{10}$$

In this case, $\widetilde{\mathbf{F}}_{\circlearrowright}$ and $\widetilde{\mathbf{F}}_{\circlearrowleft}$ maintain the same spatial orientation, but the spatial orientation of the vector potential $\widetilde{\mathbf{A}}_{\circlearrowright}$ is rotated by 180° with respect to that of $\widetilde{\mathbf{A}}_{\circlearrowleft}$.

Regardless of the definition of the bicircular field that we use, it is the orientation of the vector potential relative to the crystal lattice (or Brillouin zone) that determines the band structure modification.

### Laser-induced CNNN hopping.

To understand the physical mechanism governing the band modification and the consequent valley polarization, we will use a two-orbital tight-binding model of gapped graphene, which captures the essential physics of the problem. The gapped-graphene lattice is the same as that of hBN, with one orbital per site. Those in sublattice A have on-site energy $E_A$ whereas those in sublattice B have on-site energy $E_B$. We will assume that, in the field-free state, next-nearest neighbour hoppings $\gamma^{(\mathrm{NNN})}$ are negligible. However, upon interaction with the bicircular field, $\gamma^{(\mathrm{NNN})}$ can be induced through virtual nearest neighbour hoppings $\gamma^{(\mathrm{NN})}$. We will calculate this correction to lowest order in the field–matter interaction. To do so, note that a laser field modifies the nearest neighbour hopping between an orbital in sublattice A and an orbital in sublattice B according to the Peierls substitution,

$$\gamma_{\mathrm{AB}}^{(\mathrm{NN})}(t) = \gamma_1 e^{-i\mathbf{r}_{\mathrm{AB}}\cdot\mathbf{A}(t)}, \tag{11}$$

where $\gamma_1$ is the field-free nearest neighbour hopping, $\mathbf{r}_{\mathrm{AB}}$ is the distance from the site in sublattice B to the site in sublattice A and $\mathbf{A}(t)$ is the vector potential of the field. The laser-induced hopping term can be separated into one cycle-averaged term and one term that contains the dynamical corrections to this cycle average:

$$\gamma_{\mathrm{AB}}^{(\mathrm{NN})}(t) = \gamma_1\langle e^{-i\mathbf{r}_{\mathrm{AB}}\cdot\mathbf{A}(t)}\rangle + V_{\mathrm{AB}}(t), \tag{12}$$

where $\langle\rangle$ means cycle-averaged. Therefore, we have

$$V_{\mathrm{AB}}(t) = \gamma_1 e^{-i\mathbf{r}_{\mathrm{AB}}\cdot\mathbf{A}(t)} - \gamma_1\langle e^{-i\mathbf{r}_{\mathrm{AB}}\cdot\mathbf{A}(t)}\rangle. \tag{13}$$

This perturbation can lead to transitions between next-nearest neighbours (A ← A′) of the same sublattice through virtual nearest

neighbours. To lowest order, the transition amplitude for such a second-order process reads (atomic units are used throughout)

$$\mathcal{A}^{(2)}_{A \leftarrow A'}(t) = -\sum_B \int_{t_0}^t dt' \int_{t_0}^{t'} dt'' e^{i(E_{AB}t' + E_{BA}t'')} V_{AB}(t') V_{BA'}(t''), \quad (14)$$

where the energy difference $E_{AB} = E_A - E_B = \Delta$, where $\Delta$ is the minimum bandgap energy (we take $E_A > E_B$). The summation in principle runs along all possible intermediate states in sublattice B. However, note that only one nearest neighbour site can participate. For example, looking at Extended Data Fig. 1, we see that the transition from site $A_{-11}$ to site $A_{00}$ can happen only to first order through $B_{01}$. Therefore, we can drop the summation. Using equation (13):

$$\mathcal{A}^{(2)}_{A \leftarrow A'}(t) = -\gamma_1^2 \int_{t_0}^t dt' e^{i\Delta t'} [e^{-i\mathbf{r}_{AB}\cdot\mathbf{A}(t')} - \langle e^{-i\mathbf{r}_{AB}\cdot\mathbf{A}(t)}\rangle]$$

$$\int_{t_0}^{t'} dt'' e^{-i\Delta t''} [e^{-i\mathbf{r}_{BA'}\cdot\mathbf{A}(t'')} - \langle e^{-i\mathbf{r}_{BA'}\cdot\mathbf{A}(t)}\rangle]. \quad (15)$$

We perform a change of variables in the second integral, $u = -i[\Delta t'' + \mathbf{r}_{BA'} \cdot \mathbf{A}(t'')]$, so that $du = -i[\Delta - \mathbf{r}_{BA'} \cdot \mathbf{F}(t'')] dt''$. For moderately strong fields and large gap materials, as in this case, we have that $|\mathbf{F} \cdot \mathbf{r}_{BA}| \ll |\Delta|$, so that we can neglect the second term in $du$ and write $du = -i\Delta dt''$. In this way, the second integral is easily computed as

$$\int_{u(t_0)}^{u(t')} du \frac{e^u}{-i\Delta} - \langle e^{-i\mathbf{r}_{BA'}\cdot\mathbf{A}(t)}\rangle \int_{t_0}^{t'} dt'' e^{-i\Delta t''} = i\frac{e^{-i[\Delta t' + \mathbf{r}_{BA'}\cdot\mathbf{A}(t')]}}{\Delta}$$

$$-i\langle e^{-i\mathbf{r}_{BA'}\cdot\mathbf{A}(t)}\rangle \frac{e^{-i\Delta t'}}{\Delta} - i\frac{e^{-i\Delta t_0}}{\Delta} + i\langle e^{-i\mathbf{r}_{BA'}\cdot\mathbf{A}(t)}\rangle \frac{e^{-i\Delta t_0}}{\Delta} \quad (16)$$

$$= \frac{i}{\Delta} e^{-i\Delta t'} [e^{-i\mathbf{r}_{BA'}\cdot\mathbf{A}(t')} - \langle e^{-i\mathbf{r}_{BA'}\cdot\mathbf{A}(t)}\rangle] - \frac{i}{\Delta} e^{-i\Delta t_0} [1 - \langle e^{-i\mathbf{r}_{BA'}\cdot\mathbf{A}(t)}\rangle].$$

Substituting equation (16) into equation (15):

$$\mathcal{A}^{(2)}_{A \leftarrow A'}(t) = -\frac{i\gamma_1^2}{\Delta} \int_{t_0}^t dt' \{[e^{-i\mathbf{r}_{AB}\cdot\mathbf{A}(t')} - \langle e^{-i\mathbf{r}_{AB}\cdot\mathbf{A}(t)}\rangle]$$

$$\times [e^{-i\mathbf{r}_{BA'}\cdot\mathbf{A}(t')} - \langle e^{-i\mathbf{r}_{BA'}\cdot\mathbf{A}(t)}\rangle] - e^{-i\Delta(t_0 - t')} \quad (17)$$

$$\times [1 - \langle e^{-i\mathbf{r}_{BA'}\cdot\mathbf{A}(t)}\rangle][e^{-i\mathbf{r}_{AB}\cdot\mathbf{A}(t')} - \langle e^{-i\mathbf{r}_{AB}\cdot\mathbf{A}(t)}\rangle]\}.$$

The above expression is for the lowest-order transition amplitude between two sites in sublattice A. From it, we can identify the transition matrix element between the next-nearest neighbour sites A' and A ($\gamma^{(NNN)}_{AA'}$) by realizing that the transition amplitude is defined as

$$\mathcal{A}_{i \leftarrow k}(t) = -i \int_{t_0}^t dt' e^{i(E_i - E_k)t'} \gamma_{ik}(t'). \quad (18)$$

Hence,

$$\gamma^{(NNN)}_{AA'} = \frac{\gamma_1^2}{\Delta} \{[e^{-i\mathbf{r}_{AB}\cdot\mathbf{A}(t')} - \langle e^{-i\mathbf{r}_{AB}\cdot\mathbf{A}(t)}\rangle][e^{-i\mathbf{r}'_{BA}\cdot\mathbf{A}(t')} - \langle e^{-i\mathbf{r}_{BA'}\cdot\mathbf{A}(t)}\rangle]$$

$$-e^{-i\Delta(t_0 - t')}[1 - \langle e^{-i\mathbf{r}_{BA'}\cdot\mathbf{A}(t)}\rangle][e^{-i\mathbf{r}_{AB}\cdot\mathbf{A}(t')} - \langle e^{-i\mathbf{r}_{AB}\cdot\mathbf{A}(t)}\rangle]\}$$

$$= \frac{\gamma_1^2}{\Delta} \{e^{-i\mathbf{r}_{AA}\cdot\mathbf{A}(t')} - e^{-i\mathbf{r}_{AB}\cdot\mathbf{A}(t')}\langle e^{-i\mathbf{r}_{BA}\cdot\mathbf{A}(t)}\rangle \quad (19)$$

$$-\langle e^{-i\mathbf{r}_{AB}\cdot\mathbf{A}(t)}\rangle e^{-i\mathbf{r}_{BA}\cdot\mathbf{A}(t')} + \langle e^{-i\mathbf{r}_{AB}\cdot\mathbf{A}(t)}\rangle \langle e^{-i\mathbf{r}_{BA'}\cdot\mathbf{A}(t)}\rangle$$

$$-e^{-i\Delta(t_0 - t')}[e^{-i\mathbf{r}_{AB}\cdot\mathbf{A}(t')} - \langle e^{-i\mathbf{r}_{AB}\cdot\mathbf{A}(t)}\rangle$$

$$-e^{-i\mathbf{r}_{AB}\cdot\mathbf{A}(t')}\langle e^{-i\mathbf{r}_{BA'}\cdot\mathbf{A}(t)}\rangle + \langle e^{-i\mathbf{r}_{AB}\cdot\mathbf{A}(t')}\rangle\langle e^{-i\mathbf{r}_{BA'}\cdot\mathbf{A}(t)}\rangle]\}.$$

Averaging now over one cycle,

$$\langle\gamma^{(NNN)}_{AA'}\rangle = \frac{\gamma_1^2}{\Delta} \{\langle e^{-i\mathbf{r}_{AA}\cdot\mathbf{A}(t)}\rangle - \langle e^{-i\mathbf{r}_{AB}\cdot\mathbf{A}(t)}\rangle\langle e^{-i\mathbf{r}_{BA'}\cdot\mathbf{A}(t)}\rangle\}. \quad (20)$$

If the transition is between two next-nearest neighbours of the other sublattice (B), then we need to substitute $\Delta \to -\Delta$, and

$$\langle\gamma^{(NNN)}_{BB'}\rangle = -\langle\gamma^{(NNN)}_{AA'}\rangle. \quad (21)$$

Extended Data Fig. 2 shows some representative examples of $\langle\gamma^{(NNN)}_{BB'}\rangle$ and $\langle\gamma^{(NNN)}_{AA'}\rangle$ for different vector potentials. Note the following. First, the cycle-averaged, laser-induced, next-nearest neighbour, hoppings are complex whenever the vector potential is not pointing towards the M direction. The imaginary component is a maximum when the vector potential points along K or K', and it switches sign between these two orientations. Second, the hopping depends only on the orientation of the vector potential and not its sense of rotation (compare Extended Data Figs. 2b and 2d). The band structure is modified by rotating the vector potential, which can be achieved on a subcycle timescale by changing the two-colour phase delay $\varphi$. Finally, also note that even when the vector potential is pointing towards the point M, the bandgap is reduced relative to the field-free case due to a modification of the hopping. In this case, however, as the next-nearest neighbour hopping is real, the gap is reduced equally in both valleys.

**Numerical calculations.** We performed time-dependent simulations of a tight model of gapped graphene using the code described in ref. 44. The field-free tight-binding parameters are $r_0 = 2.73$ a.u., $\Delta = \Delta_A - \Delta_B$, where $\Delta_A = 5.9/2$ eV and $\Delta_B = -5.9/2$ eV, and $\gamma_1 = 0.089$ a.u. The atomic distance and first neighbour hopping are taken to be like those of graphene. Next-nearest neighbour hoppings and higher were neglected. Owing to the uncertainty in the experimental intensity, we simulated several ratios and intensities of the bicircular (trefoil) field. The fields were simulated using a Gaussian envelope with 30 fs of full-width at half-maximum for both fields, which matches the estimated duration of the fields in the experiment. The time-dependent propagation was converged on a Monkhorst–Pack $k$ grid of 300 × 300 points and a time step of 0.1 a.u. The dephasing time was set to 6 fs, but different values did not change our results.

Extended Data Fig. 2 shows the results for two different helicities of the bicircular field. In this case, the field parameters are $F_\omega = 2F_{2\omega} = 0.0085$ a.u., which correspond to an intensity in the crystal of $I_\omega = 2.5$ TW cm$^{-2}$ and $I_{2\omega} = 0.63$ TW cm$^{-2}$. First, note that valley polarization does not change between Extended Data Fig. 2c and Extended Data Fig. 2f, despite having switched the field helicity. Second, the valley polarization switches in Extended Data Fig. 2c and Extended Data Fig. 2f upon a 60° rotation of the bicircular field. Third, there is a not a perfect interchange of the K and K' valley populations in Extended Data Fig. 2f,i. This is because the fields (or the associated vector potential) that give rise to the populations in Extended Data Fig. 2f,i are not time-reversal partners. However, the fields in Extended Data Fig. 2c and Extended Data Fig. 2i do switch exactly the K and K' populations, since the fields in this case are time-reversal partners. Yet, it is clear that, for fixed helicity, the orientation of the field relative to the lattice controls the valley polarization, even if the switching is not fully symmetric. This effect signals band modification by the strong bicircular field and allows for subcycle control.

*k*-resolved populations. Extended Data Fig. 3 shows the $k$-resolved populations after the interaction with the bicircular field for different intensities using a fixed $\omega$ to $2\omega$ ratio of 2:1 in intensity. We found similar results for intensity ratios of 1:1, 4:1 and 6:1. The polarized valley always corresponds to that in which the model predicts that the bandgap is reduced. Also, the valley polarization increases as a function of intensity, in accordance with the model prediction that the effective bandgap decreases as the intensity is increased.

**Probing the valley polarization.** To transfer these valley populations into an optical degree of freedom that can be measured in the experiment, we used a linearly polarized probe pulse after the bicircular pulse which, as explained in the main text, allowed us to map them into the

helicity of its nonlinear harmonic response (H3 in this case). As in the experiment, we used a probe light field of the same $\omega$ frequency as the fundamental field in the bicircular pulse. The field strength of the probe field was $F_{probe} = 0.01$ a.u., it was polarized along the $\Gamma$–M direction ($x$ direction in the figures) and its duration was 30 fs of full-width at half-maximum. We tested different probe field strengths, but these did not affect our results.

Extended Data Fig. 4 shows the results of the polarimetry analysis. The two helical components that the two photodiodes measure are defined as

$$\hat{e}_l = \frac{1}{\sqrt{2}}(\hat{e}_x + i\hat{e}_y),$$
$$\hat{e}_r = \frac{1}{\sqrt{2}}(\hat{e}_x - i\hat{e}_y).$$
(22)

Therefore,

$$\hat{e}_l = \left| \frac{A_x}{\sqrt{2}}e^{i\phi_x} + \frac{A_y}{\sqrt{2}}e^{i(\phi_y + \pi/2)} \right|^2,$$
$$\hat{e}_r = \left| \frac{A_x}{\sqrt{2}}e^{i\phi_x} + \frac{A_y}{\sqrt{2}}e^{i(\phi_y - \pi/2)} \right|^2,$$
(23)

where $A_x$ and $A_y$ ($\phi_x$ and $\phi_y$) are the Fourier amplitudes (phases) of the harmonic of interest of the probe (H3 in this case), along directions $x$ and $y$, respectively. We can rewrite the above as

$$\hat{e}_l = \frac{1}{2}|A_x| + \frac{1}{2}|A_y| + |A_x||A_y|\cos(\phi_x - \phi_y - \pi/2),$$
$$\hat{e}_r = \frac{1}{2}|A_x| + \frac{1}{2}|A_y| + |A_x||A_y|\cos(\phi_x - \phi_y + \pi/2).$$
(24)

The phase $\phi_y$ changes by $\pi$ as the valley polarization changes sign since the $y$ component of the current comes from the anomalous contribution and, thus, is proportional to the Berry curvature, which has the same magnitude but opposite sign in both valleys. As the valley polarization changes upon rotation of the bicircular field, the interference (cosine) term in the expression above oscillates sinusoidally, switching sign with a switch in valley polarization. Additionally, the interference term of the two helicities oscillates out of phase because of the factor $\pm\pi/2$. Extended Data Fig. 4 shows the interference term of the two helicity components, which indeed oscillate sinusoidally and out of phase. As expected, these interference terms are a maximum or a minimum when there is maximum valley polarization and zero when there is none. The asymmetry of these two interference terms completely characterizes the valley polarization.

Yet, the signal observed in the experiment also includes the amplitude term in the equation above, $\frac{1}{2}|A_x| + \frac{1}{2}|A_y|$, which also oscillates. Its oscillation comes from unequal population injection as a function of rotation and other nonlinear effects occurring during the harmonic generation. However, this term is the same for both helicities, and thus, it is merely a background introducing higher-order Fourier components into the oscillation. We plot this term in Extended Data Fig. 4.

The helicity signal is then the sum of the amplitude term, which is common to both helicities, and the interference term, which is different for each helicity and which contains the information on the valley polarization. The total left and right signals give, respectively, the red and green curves in Extended Data Fig. 4 and also in the main text, which is the curve to be compared with the experiment.

Importantly, regardless of the value of the amplitude term, since it is a background common to both helicities, we can remove its influence. For this, we Fourier-filtered the oscillation to extract only the 120° periodic oscillation. In this way, the asymmetry of the two helicity signals after Fourier filtering characterizes the valley polarization, with the change of sign indicating valley switching.

Note that the helicity-resolved H3 probe signal as a function of the pump trefoil rotation is essentially the same regardless of whether the probe is polarized along $\Gamma$–K or $\Gamma$–M (Extended Data Fig. 5).

## Experimental details

**Laser system.** The laser pulses used for the experiments were from a mid-infrared laser system based on optical parametric chirped-pulse amplification (OPCPA). The details are in ref. 45. In brief, the laser for the mid-infrared OPCPA was from a 1 µm Innoslab Yb laser system. The pulses were used to generate the seed and the pump for the OPCPA. The OPCPA system produced pulses at 2.03 µm centre wavelength with a 100 kHz repetition rate. During the experiment, the average power obtained from the laser was 5.5–6 W with a pulse duration of approximately 25 fs. The pulse duration was carefully characterized using a frequency-resolved optical gating[46] with a set-up developed in-house. The laser pulses were characterized after they had passed through the interferometer. To accurately determine the temporal characteristics of the pulses used in the experiment, they were picked up right before they entered the reflective objective of the microscope arrangement. A second-harmonic generation (SHG) frequency-resolved optical gating was used to characterize the 1 µm pulses in a beta-barium borate crystal, whereas the 2 µm pulses were characterized by a surface third-harmonic generation frequency-resolved optical gating on glass (NBK7), both in a non-collinear geometry. The results of these measurements are shown below.

**Interferometer.** A large part of the experimental set-up was a three-arm Mach–Zehnder interferometer, as shown in Extended Data Fig. 6. Various aspects of the interferometer are described in detail in the following sections.

**Pump.** The 2 µm wavelength pulses from the laser source were fed directly into the three-arm interferometer. First, the pulses were split in two by a 50–50 beam splitter (red beam from right to left in Extended Data Fig. 6). The reflected part proceeded to the pump and the transmitted one to the probe arm. The pump beam was sent through a 1.5-mm-thick lithium niobate (LiNbO₃) crystal (SHG) cut at 45.6°. This interaction produced a second pulse at half the wavelength (frequency of $2\omega$) of the fundamental (frequency of $\omega$) by sum frequency generation (SHG)[47]. LiNbO₃ is suited well for a 2 µm fundamental wavelength, given its wide transparency range in the mid-infrared of up to 3.5 µm. The phase-matching conditions (type 1) led to the second-harmonic component being cross-polarized with respect to the fundamental pulse[47]. The co-propagating two-colour pulses ($\omega$ and $2\omega$) were separated using a custom-designed dichroic beam splitter. The $\omega$ arm passed through a variable neutral-density filter, a retro-reflection stage and CaF₂ as used in the probe arm, but without the piezo motion rails, as the length of this arm was fixed with respect to the other arms. Additionally, a wire grid polarizer was placed in its path to clean its polarization state before it combined with the $2\omega$ arm at another dichroic beam splitter. The $2\omega$ arm was reflected off the first dichroic beam splitter at 90° and traversed the exact same arrangement as the $\omega$ arm. The only difference was a different pair of custom chirped mirrors for $2\omega$ on the delay stage above a closed-loop stick–slip piezo nano-positioning rail with a positioning resolution of 1 nm, a different amount of dispersion-compensating material and a perpendicular axis set on the polarizer. The pump arm (co-propagating $\omega$ and $2\omega$) was further sent through a long-pass spectral filter (F1) to block the parametric optical signals generated at the harmonics of the two-colour pump in the LiNbO₃ crystal. Finally, the two-colour pump ($\omega$ and $2\omega$) pulses passed through the fused-silica wedged pair to recombine with the probe pulses ($\omega$), which were reflected off it.

After the three-arm interferometer, the laser beam was expanded using a reflective telescope arrangement to roughly match the input

diameter of the tight-focusing reflective objective and adjust the beam divergence. However, right before the focusing objective, there was a broadband quarter-wave plate ($\lambda$/4) with its optical axis at 45° with respect to the cross-polarized axis of the two-colour pump. This wave plate arrangement transformed the linear, cross-polarized, pump pulses into circularly polarized, counter-rotating pulses that look like a trefoil or a three-leaf pulse in the $X$–$Y$ plane, as required in the experiment. The quarter-wave plate was intentionally placed right before the focusing element to prevent any changes in ellipticity due to a phase shift between the s- and p-polarized components upon reflection, especially from beam-folding mirrors and other coated optical elements. For the $\omega$ and $2\omega$ pump components, the pulse durations measured just before the focusing objective were about 26 and 48 fs, respectively. Extended Data Fig. 7 depicts the spectral and temporal characteristics of the $\omega$ and $2\omega$ pump components. A similar kind of waveform synthesizer was also used earlier in attosecond-controlled strong-field experiments by the group[36]. The data shown in the manuscript were obtained with the pump intensity in the range 4–7 TW cm$^{-2}$. For the intensity scaling measurements, the overall power of the pump beam was changed with a variable neutral-density filter keeping the relative power ratio between the components intact.

**Probe.** In the probe arm, the pulses went through a variable neutral-density filter followed by a delay stage, which, along with a pair of silver retro-reflecting mirrors, hosted two customized chirped mirrors for simultaneous positive dispersion and spectral filtering. Spectral filtering is crucial given the existence of weak optical signals at lower wavelengths arising from the parametric processes in various crystals in the laser system. The delay stage was mounted on a closed-loop stick–slip piezo rail (Smaract SLC-24 series) with a positioning resolution of 1 nm. Further down, the pulses went through a defined thickness of material ($CaF_2$) to compensate for excess positive dispersion. Additionally, another long-pass spectral filter was placed in the beam path to further suppress the unwanted optical signals at lower wavelengths. Finally, the probe pulses went through a half-wave ($\lambda$/2) plate and a quarter-wave ($\lambda$/4) plate, after which they were reflected off the wedge plate and recombined with the pump beam. This wave-plate combination allowed us to control the polarization state of the probe pulses, and the mechanism is described in greater detail later. This wedge pair arrangement not only acted as a beam recombination element but also as a power attenuator for the probe pulses, as only 4% of the power was reflected. After being recombined, the probe beam followed the collinear path with the pump beam. Just before the focusing objective, a pulse duration of about 26 fs was achieved for the probe pulses.

As shown in Extended Data Fig. 6, a $\lambda$/4 plate was the last optical element before the pulses entered the reflective objective. This led to a major problem in which the third arm (or the linearly polarized probe as in this experiment) cannot remain linear once it has passed through the $\lambda$/4 plate unless it is along the optical axis at 45°. Also, a linear polarization launched at 45° with respect to the s- or p-polarized states would lose its linear contrast, as it would become elliptical on acquiring a different phase shift in the s and p components on every reflective optic in its beam path. To have full flexibility over the polarization state of the probe pulses after the $\lambda$/4 plate, a scheme was implemented such that a combination of $\lambda$/2 and $\lambda$/4 wave plates were additionally placed in the probe arm, as depicted in Extended Data Fig. 6. Intuitively, one can think of these additional plates as inducing perfectly opposing elliptical polarization, which cancels out in the final $\lambda$/4 before the reflective objective to produce linearly polarized light with a high extinction ratio. This scheme was numerically tested using a Jones matrix approach. It was observed that any arbitrary shifts in phase between the s and p components in the beam path between the two $\lambda$/4 wave plates can be compensated. However, changes in magnitude between the s and p components lead to a deviation from linearity and cannot be compensated for by this scheme.

A movable silver mirror was placed at 45° right before the reflective objective intercepting the probe pulses to optimize and characterize the polarization extinction ratio or linearity. The pulses were then guided to an InGaAs photodiode with a polarizer attached to it at a fixed angle. The fixed angle was such that the probe polarization was aligned along s or p to prevent any additional ellipticity induced by the intercepting mirror, which would not be present otherwise during the experiment.

The data shown in the main manuscript were obtained at a pump–probe delay of about 60–110 fs.

**Trefoil pump rotation.** When the bicircular $\omega$ and $2\omega$ components of the pump were combined in the interferometer such that the $E$ field ratio at the focus was 2:1, the coherent sum of their electric fields in the $X$–$Y$ plane (the plane perpendicular to propagation) was transformed to that of a trefoil waveform. The rotation of the trefoil waveform was controlled by the subcycle phase delay between the $\omega$ and $2\omega$ components. Experimentally, this was achieved by introducing an optical path difference between the $\omega$ and $2\omega$ arms. When the central wavelength ($\lambda_0$) of the $\omega$ arm was 2 μm, a rotation of 360° was induced by delaying the piezo stage by 3 μm. This information was used to convert the stage delay to angular rotation, which was recorded as the experimental data.

**Interferometric stability.** During the experiment, it was critical that the angle of the three-leaf or trefoil pump remained stable, as this was directly linked to the delay between the two colours in the pump arm. To characterize the delay stability, an additional temperature-stabilized continuous-wave diode laser (Thorlabs L785P090 with LDMT9) was sent through the two pump arms to interferometrically measure its path difference over time.

Using the above-mentioned scheme, the interferometric stability was found to be highest around 2 h after switching on the driving laser system. Additionally, tests were carried out to measure the stability when the laser was going through the interferometer and the piezo delay stage being scanned, as during the experiment, as illustrated in Extended Data Fig. 8. Over a period of 10 min, the standard deviation of the position generated by the closed-loop piezo stages from the position extracted from the continuous-wave interferometer was close to 38.1 nm, which roughly translates to about 4.6° in rotation error of the $\omega$ and $2\omega$ bicircular trefoil structure. A similar scheme with active stabilization was used earlier and achieved few tens of attosecond interferometric stability[48].

## Microscope

The core segment of the microscope (Extended Data Fig. 9) applied tight focusing of laser pulses from all three interferometer arms, followed by a collection and collimation configuration before the light proceeded to the detection apparatus. A 500-μm-thick fused-silica substrate with the hBN monolayer on its front surface was placed between these components.

The focusing was achieved without adding any dispersion or chromatic aberration by using a reflective objective with a numerical aperture of 0.4. The beam profile in the interaction region was characterized by placing a CMOS sensor in the focal plane. A cross section of the resulting two-dimensional signal on the sensor is shown in Extended Data Fig. 10 for all three interferometer arms. The beam waists extracted from the fit were about 8.0, 10.0 and 9.7 μm for the $2\omega$ pump, $\omega$ pump and $\omega$ probe arms, respectively. The beam was largely circular at the focus, and the residual ellipticity was below 5% for each arm. The small focus spot sizes not only allowed us to spatially restrict the laser interaction to a single hBN monocrystalline grain but also gave a Rayleigh length (201 μm for $2\omega$ and 157 μm for $\omega$) that was significantly less than the thickness of the fused-silica substrate (Extended Data Fig. 9). By observing the surface-enhanced perturbative harmonics, we could individually optimize the substrate position (back or front of the hBN-coated surfaces) at the focus.

Once the fused-silica substrate was at the focus, the diverging beam along with other nonlinear signals was recollimated using a long-working-distance transmissive objective with a numerical aperture of 0.45. Using a closely matched numerical aperture allowed the divergent beams to be entirely collected.

Further down, the laser pulses were polarization resolved using a quartz Wollaston prism. The prism introduced an angular separation of 10° between the p- and s-polarized components, which were then loosely focused by a $CaF_2$ lens onto the two separate large-area photodiodes. A $\lambda/4$ wave plate was placed between the transmission objective and the Wollaston prism with its optical axis at 45° with respect to the prism to convert the photodiodes into helicity detectors.

## Spatio-temporal overlap

As described earlier, the implementation of a three-arm interferometer leads to the laser pulses travelling through three different non-collinear optical paths. Upon recombination, they require careful alignment such that they spatially and temporally overlap in the interaction region. To ensure this, the second-order and third-order nonlinearity in fused silica is utilized such that a two- and three-photon transition of $\omega$ and $2\omega$ leads to the production of $3\omega$ and $4\omega$ photons, respectively. To utilize this effect, the $2\omega$ pump is chosen as a common reference and the other pulses are aligned onto this particular arm. An initial spatial alignment of all three arms using irises is enough to ensure a coarse spatial overlap between the pulses at the focus on the fused-silica substrate. Soon after, the $2\omega$ pump arm is delayed with respect to the $\omega$ pump arm while observing the $4\omega$ light with an appropriate band-pass filter before the photodiodes. As the pump arms are counter-rotating bicircularly in their polarization state with respect to each other, the $3\omega$ channel is disallowed. After the $2\omega$ pump delay is fixed, its delay with respect to the probe arm is determined by scanning the probe delay stage and observing the $3\omega$ or $4\omega$ channels, as both these transitions are allowed. After the temporal alignment, the spatial overlap is checked and optimized based on the respective parametric signals.

## Lock-in polarization detection

In this experiment, the signal of interest is the $3\omega$ harmonic generated by the probe and modulated by the pump. However, several sources of light are co-propagating along with the signal and impinging on the diodes. Hence, a simple amplified detection of the diode current is not enough to distinguish the relevant $3\omega$ signal ($\lambda \approx 667$ nm). Signals that are spectrally different are already filtered using high- or low-pass spectral filters, which block the fundamental $2\omega$, components above $\lambda = 750$ nm and other components below $\lambda = 600$ nm. However, $3\omega$ light can be produced by the pump ($\omega$) and probe ($\omega$) pulses individually by the third-harmonic upconversion process and by the combination of pump ($2\omega$) and probe ($\omega$) through a two-photon process on the silica substrate, thereby further polluting the experimental signal of interest.

We overcome this major problem by modulating either the $\omega$ arm or the $2\omega$ arm of the interferometer using a mechanical chopper wheel with a 50% duty cycle at a known frequency. Hence, the $3\omega$ signal of interest, which was probed by a modulated bicircular pump, was also modulated at the chopping frequency. To chop the $\omega$ arm, an unwanted $3\omega$ signal, generated by the third-harmonic upconversion of $\omega$ pump, falls entirely on a single diode (D1) because its helicity remains the same as that of the $\omega$ pump. Therefore, the other diode (D2) remains background-free. Moreover, when the $2\omega$ arm is chopped, an unwanted $3\omega$ signal, generated through a pump ($2\omega$) and probe ($\omega$) two-photon process, falls on the D2 diode, as its helicity is the same as that of the $2\omega$ signal. In that case, the D1 diode remains background-free. Therefore, a combination of two independent measurements while keeping all the parameters unchanged, first chopping the $\omega$ arm and subsequently the $2\omega$ arm, can fulfil the requirement for detecting both the helicities of the $3\omega$ signal in a background-free condition. Note that the continuous service of the continuous-wave reference interferometer ensures the interferometric stability between these two measurements. This modulated signal impinging on the photodiodes is amplified by two high-gain, low-noise, trans-impedance amplifiers placed right behind the respective photodiodes to minimize the noise picked up by the cables. The amplified voltage signals are then fed into two time-synchronized lock-in amplifiers (Zurich Instruments).

Extended Data Table 1 lists the signals detected along with their origin. It also depicts the method by which the background signals are rejected while keeping the relevant signal detectable.

## hBN monolayer sample

The hBN monolayer samples used in this experiment were obtained from a commercial supplier (ACS Materials). These were grown using chemical vapour deposition and subsequently transferred onto a 500-μm-thick, large-area (5 cm diameter), fused-silica substrate. Although the monolayer patches, of characteristic size of about 10 μm as typical for samples grown by chemical vapour deposition, had their large areas covered, their orientations were random. These individual hBN patches were characterized in situ by microscopy.

## SHG polarization-resolved microscopy

To characterize the quality of the hBN patches individually, we used in situ polarization-resolved SHG microscopy. This was repeated before every experimental run.

SHG polarization-resolved microscopy was done with the 1 μm wavelength arm of the set-up. The perturbative second harmonic was detected by a photodiode or spectrometer after being collected by the transmission objective and passed through band-pass filters and eventually a polarizer. The SHG is not only the strongest component but being an even harmonic, it is a pure indicator of broken inversion symmetry in a crystal. Hence, the emission was also fully absent from the fused-silica surface, leaving only odd layers of hBN ($1L$, $3L$, $5L$, ..., $(2n+1)L$), which break inversion symmetry to produce SHG. As $n$ increases, the magnitude of SHG drops[49,50]. Hence, a spatial scan over the substrate with the SHG strength as an optimization parameter can locate the thinner odd layers of hBN.

Another important advantage of this technique is that it can be performed in situ during the final band engineering experiment to make sure that all three probe and bicircular pump pulses also excite the same hBN grain as independently observed by this characterization technique.

Further, not only can this technique determine the inversion-symmetry breaking but it can also detect the orientation of the hBN crystals by observing the polarization response of the material. The two unique light-induced oscillation directions (armchair and zigzag) make the outgoing SHG polarization rotate in the opposite direction and at twice the rate, as a function of the rotation of the incoming (pump) light polarization. This has also been widely investigated and demonstrated in recent works[49,50]. Two results from two different positions on the hBN sample used are displayed in Extended Data Fig. 11. The characteristic four-lobe polarization-dependent signal implies that the interaction region of the laser irradiation is small enough to fit into the monocrystalline hBN patch. Thus, the single crystalline orientation is dominant and is not averaged out over several grains with different crystalline orientations. Further, Extended Data Fig. 11a,b shows that the four-lobe structure exhibits an angular offset with the four-lobe structure obtained from a neighbouring hBN patch. This manifests from the hBN crystalline patches, which are oriented at arbitrary angles at different laser excitation sites. Usually, a six-lobe feature in the SHG signals can be used to indicate the orientation of hBN (ref. 49). This difference is attributed to a lab-to-incident polarization frame conversion. In the present case, the SHG polarization microscopy was done in transmission mode while keeping the sample fixed for a given hBN patch. The sensitivity of this technique to the presence of hBN on the

fused-silica substrate allows it to be used as a marker for laser-induced damage. This confirms that the suitable intensity range before damage occurs is close to $10^{13}$ W cm$^{-2}$.

## Data availability
The data that support the findings of this study are available from the corresponding author upon reasonable request.

## Code availability
The parts of the numerical code used for the simulations contained in this study is available from the corresponding authors upon reasonable request. Details of the numerical code can be found in ref. 44.

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

**Acknowledgements** S.B. and M.F.K.'s work at SLAC is supported by the US Department of Energy, Office of Science, Basic Energy Sciences, Scientific User Facilities Division, and by the Chemical Sciences, Geosciences, and Biosciences division under award DE-SC0063. S.B. acknowledges a fellowship from the Alexander von Humboldt Foundation. S.M. acknowledges support from the Max Planck Society. A.J.G. acknowledges funding from the Comunidad de Madrid through TALENTO Grant 2022-T1/IND-24102 and from the EU Horizon 2020 programme (Grant Agreement 899794). R.E.F.S. acknowledges support from a fellowship (LCF/BQ/PR21/11840008) from La Caixa Foundation (Grant Nos. ID 100010434 and PID2021-122769NB-I00) funded by the Ministry for Science and Innovation (Grant No. MCIN/AEI/10.13039/501100011033). Fruitful discussions with M. Ivanov and O. Smirnova are gratefully acknowledged.

**Author contributions** S.B. conceptualized, initiated and designed the project in discussion with A.J.G., which was supervised by S.B. and M.F.K. S.B. devised the experimental methodology and design. S.M. and S.B. developed the experimental set-up, for which M.F.K. provided the basic resources. S.M., M.A. and S.B. performed the measurements. S.M. and S.B. analysed the data. S.M., A.J.G. and S.B. interpreted the results. A.J.G. and R.E.F.S. developed the theoretical model. M.N. supported the laser operations. V.P. fabricated the specialized chirped and dielectric mirrors following the requirements designed by S.M. and S.B. A.J.G. and S.B. wrote the manuscript and managed the review process with input from all co-authors.

**Competing interests** The authors declare no competing interests.

**Additional information**
**Correspondence and requests for materials** should be addressed to Álvaro Jiménez-Galán or Shubhadeep Biswas.

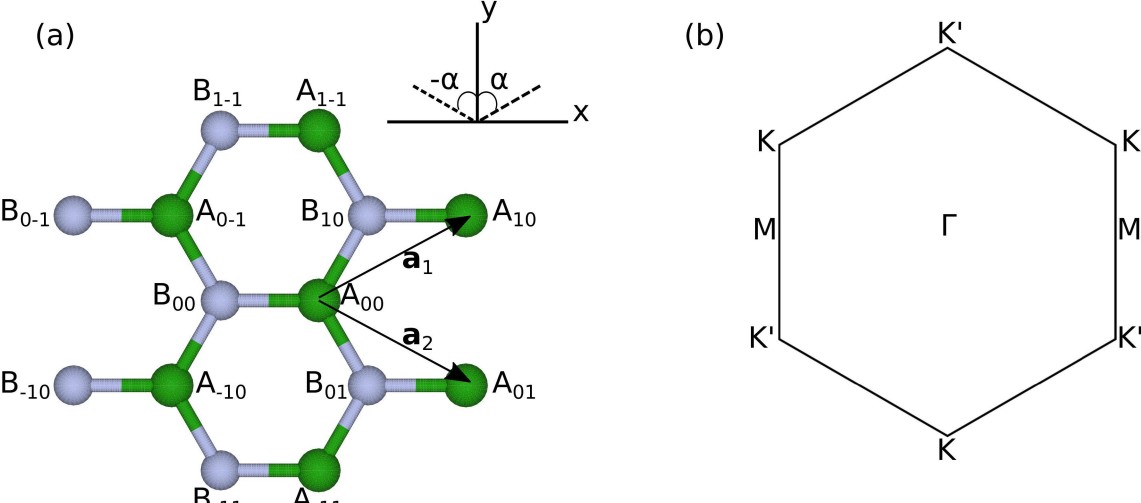

**Extended Data Fig. 1 | Crystal structure and Brillouin zone of hBN. a**, Crystal lattice of monolayer hBN up to next-nearest neighbours. Each atom $A_{nm}$ is labelled by its sub-lattice (A or B) and by its lattice vector $R_{nm} = na_1 + ma_2$. In Eq. 2, the angle $\alpha$ between two atoms is defined according to the inset in the top left. **b**, First Brillouin zone of monolayer hBN and its high-symmetry points.

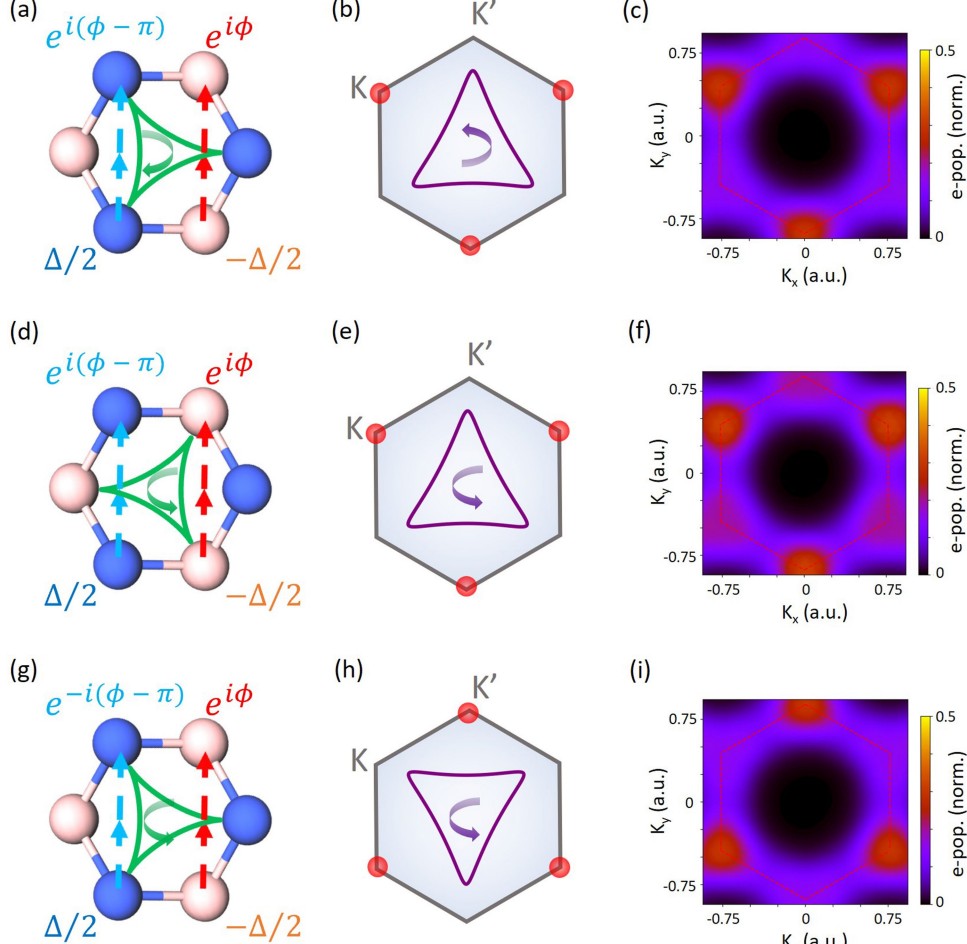

**Extended Data Fig. 2 | Laser-induced Haldane-type hoppings and band structure modification. a,d,g,** Examples of $\langle\gamma_{AA'}^{(NNN)}\rangle$ (red) and $\langle\gamma_{BB'}^{(NNN)}\rangle$ (blue) for different vector potentials. The green curve shows the electric field orientation and the arrow shows the sense of rotation of the field. **b,e,h,** The purple curve shows the vector potential that results from the electric field on the left panel, with the arrow showing its sense of rotation. The red circles show the valley at which the band gap is reduced according to the analytical model. **c,f,i,** $k$-resolved populations obtained from a time-dependent simulation of a tight binding model of gapped graphene using the electric field and vector potential in the corresponding left panels ($I_\omega = 2.5$ TW/cm$^2$, $I_{2\omega} = 0.63$ TW/cm$^2$, and frequency $\omega = 0.6$ eV).

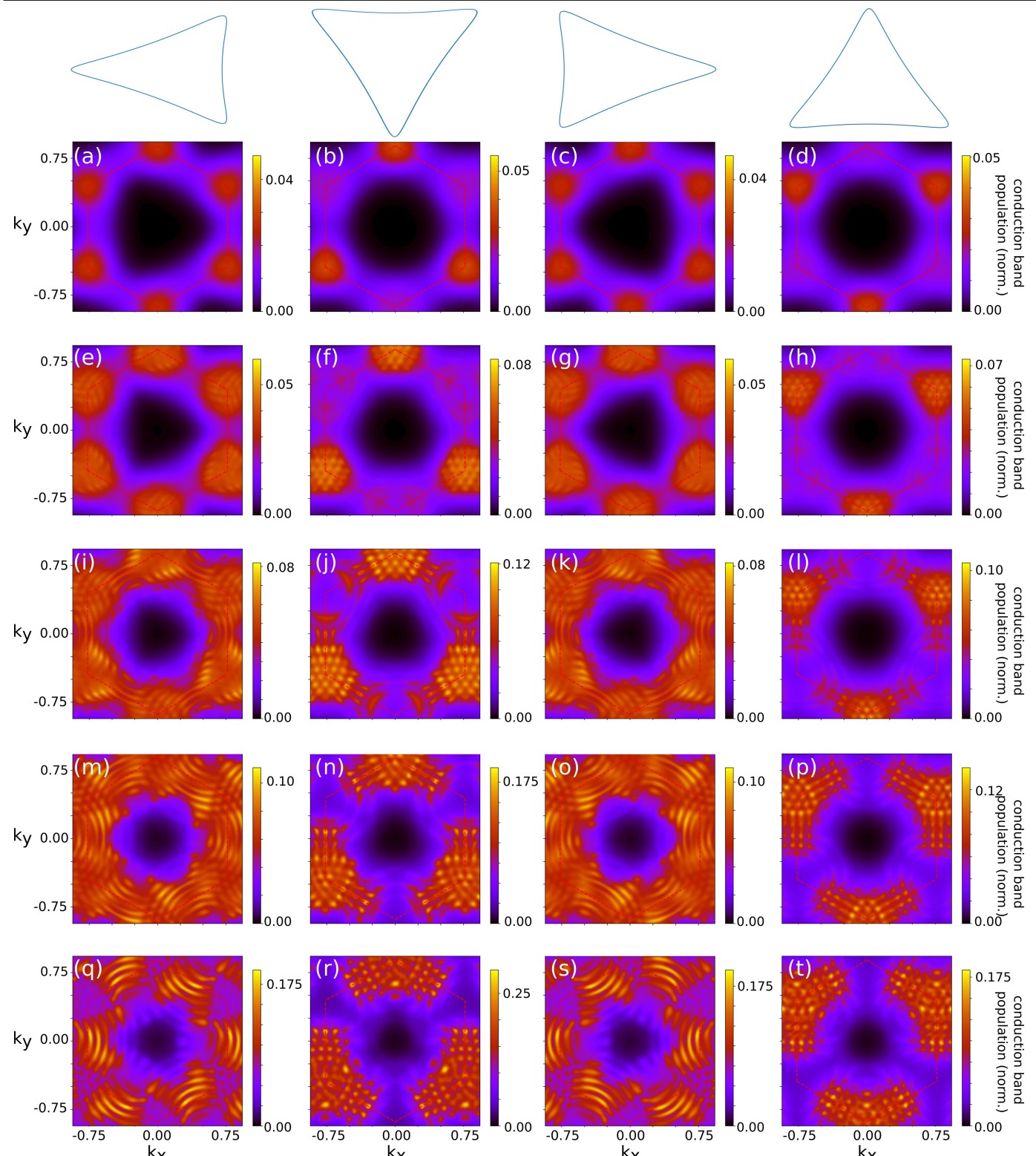

**Extended Data Fig. 3 | k-resolved populations for different pump intensities.** Normalized electron populations after the pump bicircular field for several orientations and pump intensities. The ω to 2ω intensity ratio is kept fixed at 2:1. **a–d**, $I_\omega = 2$ TW/cm²; **e–h**, $I_\omega = 4$ TW/cm²; **i–l**, $I_\omega = 6$ TW/cm²; **m–p**, $I_\omega = 8$ TW/cm²; **q–t**, $I_\omega = 10$ TW/cm². Each of the four columns corresponds to a different orientation of the vector potential relative to the Brillouin zone hexagon, which is shown at the top.

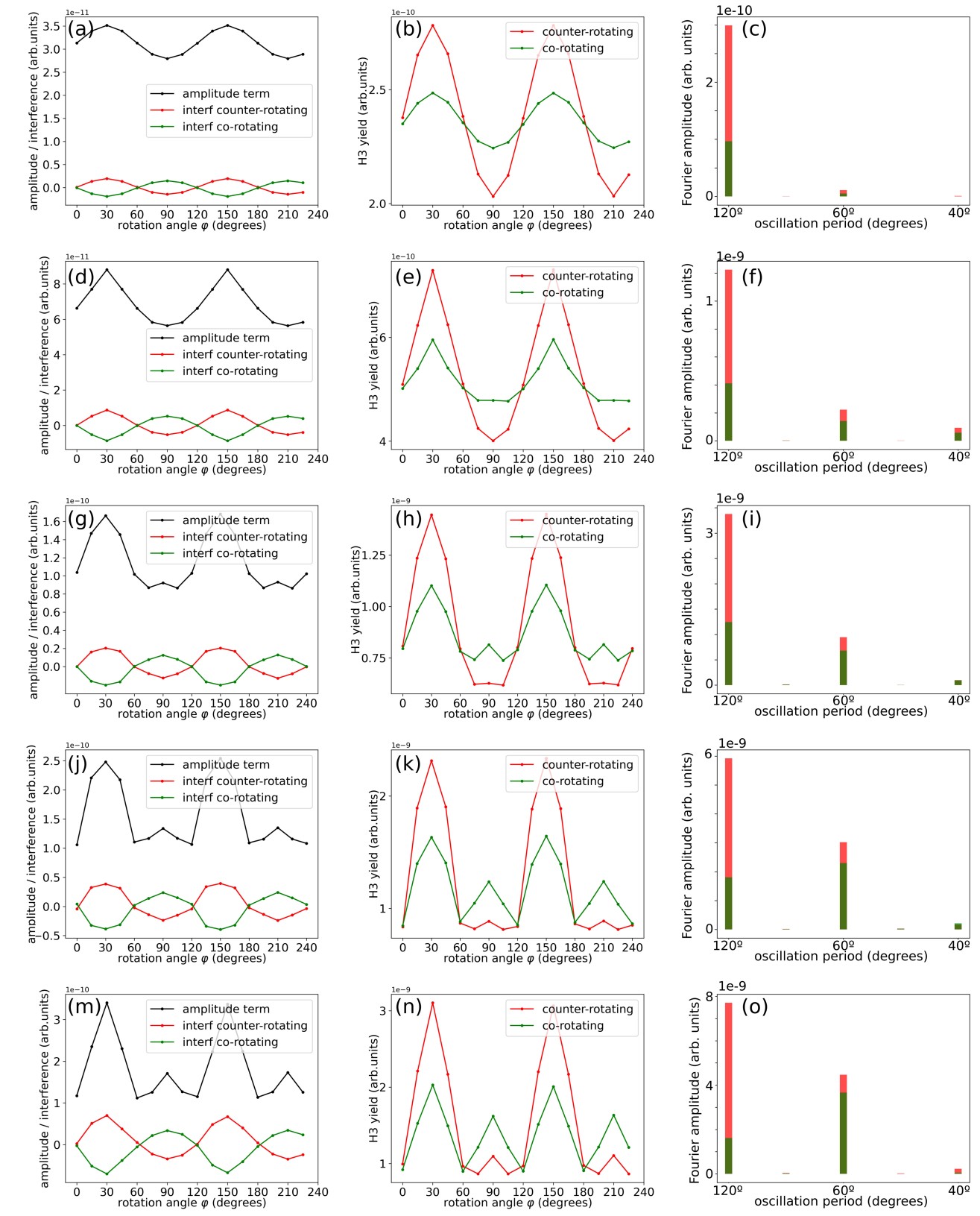

**Extended Data Fig. 4 | Probing of the valley polarization via optical polarimetry.** Left (red, counter-rotating with the fundamental ω field of the pump) and right (green, co-rotating with the fundamental ω field of the pump) helicities of the third harmonic of a linearly-polarized probe field. The different rows correspond to different intensities of the pump field keeping the ω to 2ω intensity ratio is kept fixed at 2:1 (same as those in Extended Data Fig. 3): **a–c**, $I_\omega$ = 2 TW/cm²; **d–f**, $I_\omega$ = 4 TW/cm²; **g–i**, $I_\omega$ = 6 TW/cm²; **j–l**, $I_\omega$ = 8 TW/cm²;

**m–o**, $I_\omega$ = 10 TW/cm². The probe polarization is oriented along $\Gamma - M$. The left column corresponds to the amplitude and interference signals, as discussed in Methods 1.4.2, computed at the frequency point $\nu = 3\omega$. The middle column shows the left and right helicity signals computed from the integral of the H3 signal, as measured in the experiment. The right-most column shows the Fourier components of the oscillation in the corresponding middle column panel.

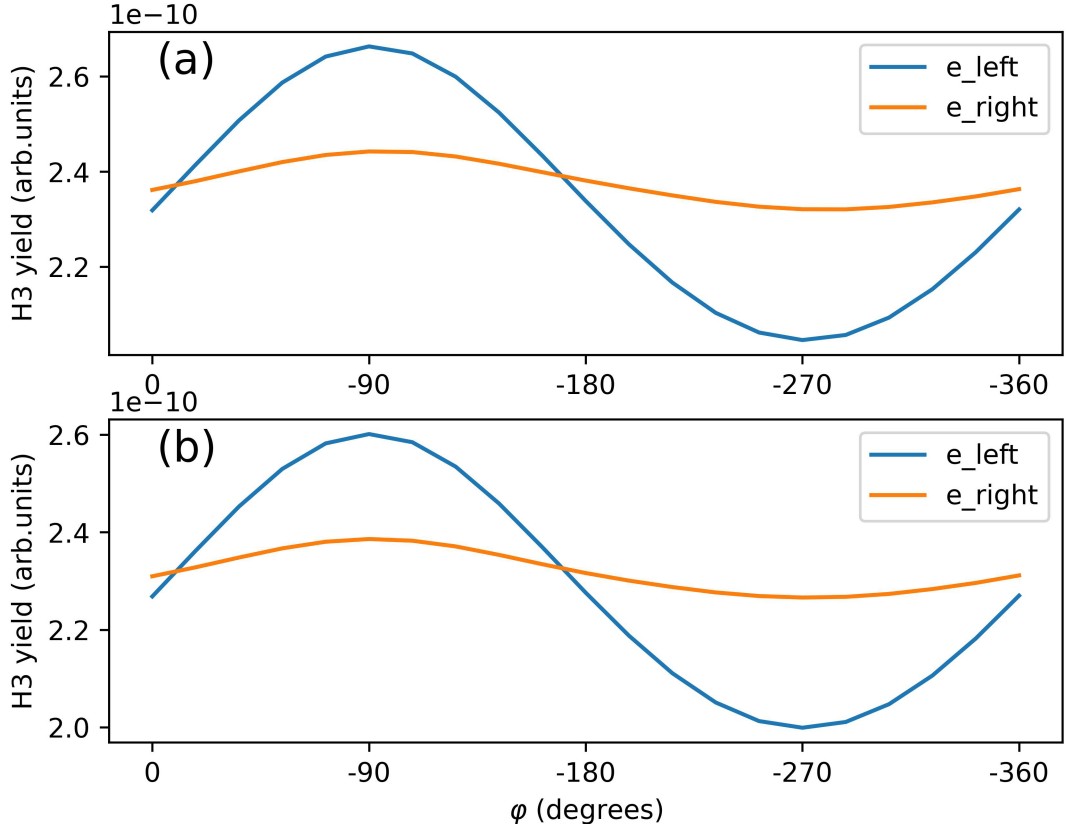

**Extended Data Fig. 5 | Dependence of helicity-resolved H3 on probe polarization direction.** Results for the helicity-resolved H3 probe signal as a function of the two-colour phase delay φ (that is, pump trefoil rotation) when the probe field is polarized along the (**a**) $\Gamma - K$ and (**b**) $\Gamma - M$ direction. The intensity of the pump field is 2 TW/cm$^2$.

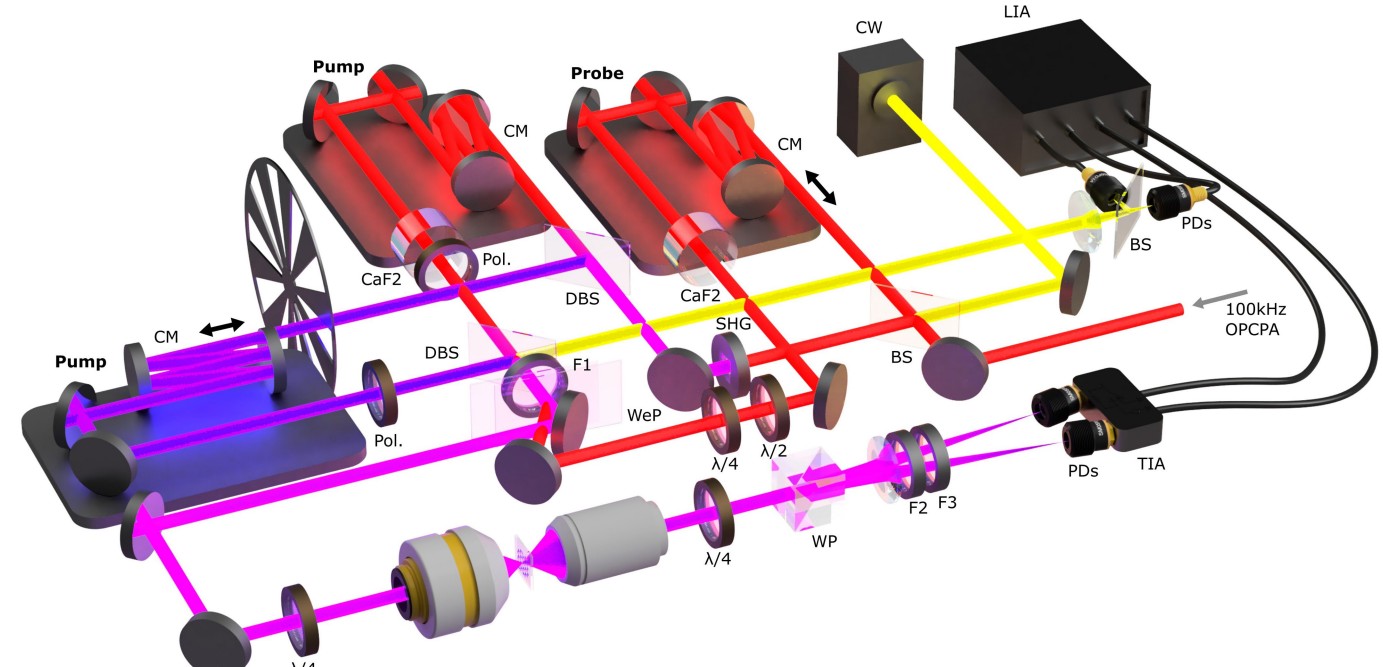

**Extended Data Fig. 6 | Mach–Zehnder interferometric setup.** The acronyms of the components have the following meaning: WP, Wollaston prism; F1, long pass filter; F2, short pass filter; F3, long pass filter; PD, photodiode; TIA, trans-impedance amplifier; BS, beam splitter; DBS, dichroic beam splitter; Pol., polarizer; WeP, wedged pair; CM, chirped mirrors; SHG, second harmonic generation LiNbO$_3$ crystal; CW, continuous wave laser; LIA, lock-in amplifier. A detailed description of the setup is provided in the text.

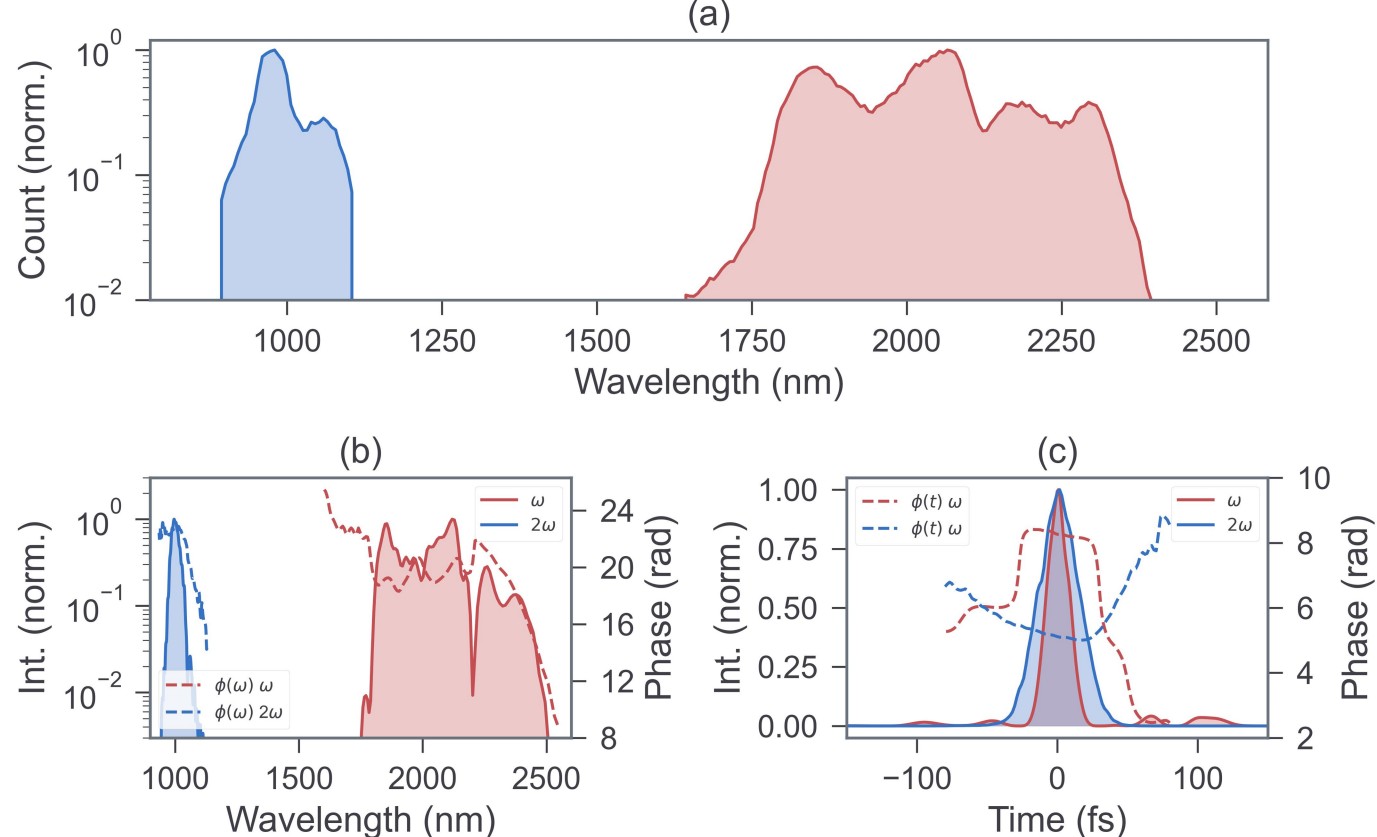

**Extended Data Fig. 7 | Spectral and temporal properties of the ω – 2ω trefoil pump pulses.** Subplot **a** shows the measured laser spectra recorded after the interferometer, right before the focusing objective to give an accurate representation of the laser pulses used. The laser pulses are picked off from the same region for frequency-resolved optical gating (FROG) retrieval. The retrieved spectra and temporal profile along with their phases are shown in subplots **b**,**c**, respectively. It suggests the pulse durations of about 26.1 and 48.0 fs for ω and 2ω, respectively.

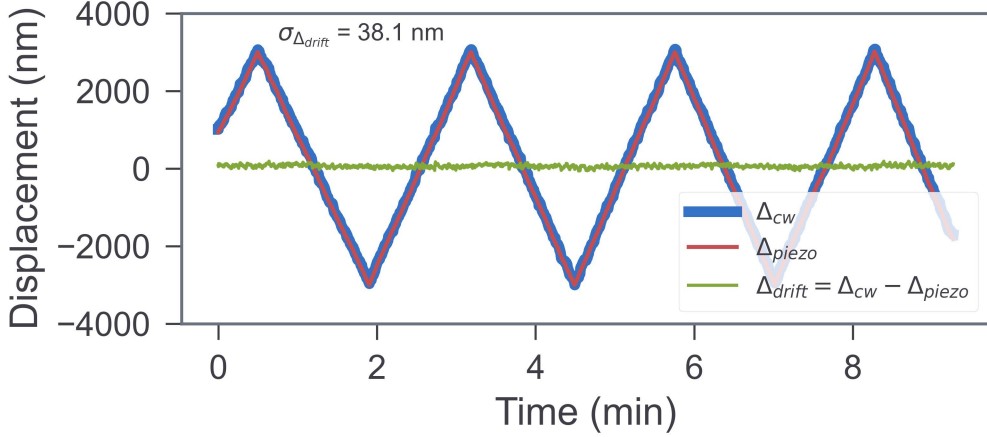

**Extended Data Fig. 8 | Interferometric stability of the pump arms.** The relative displacement between the $\omega - 2\omega$ pump arms is recorded while a piezo closed-loop stage controlling the displacement is swept over a period of ten minutes. The position read out by the stage is compared to that extracted from an additional CW interferometer to estimate the error in displacement mapping. A standard deviation in drift of 38.1 nm corresponds to an angular jitter of 4.6° in the trefoil pump structure.

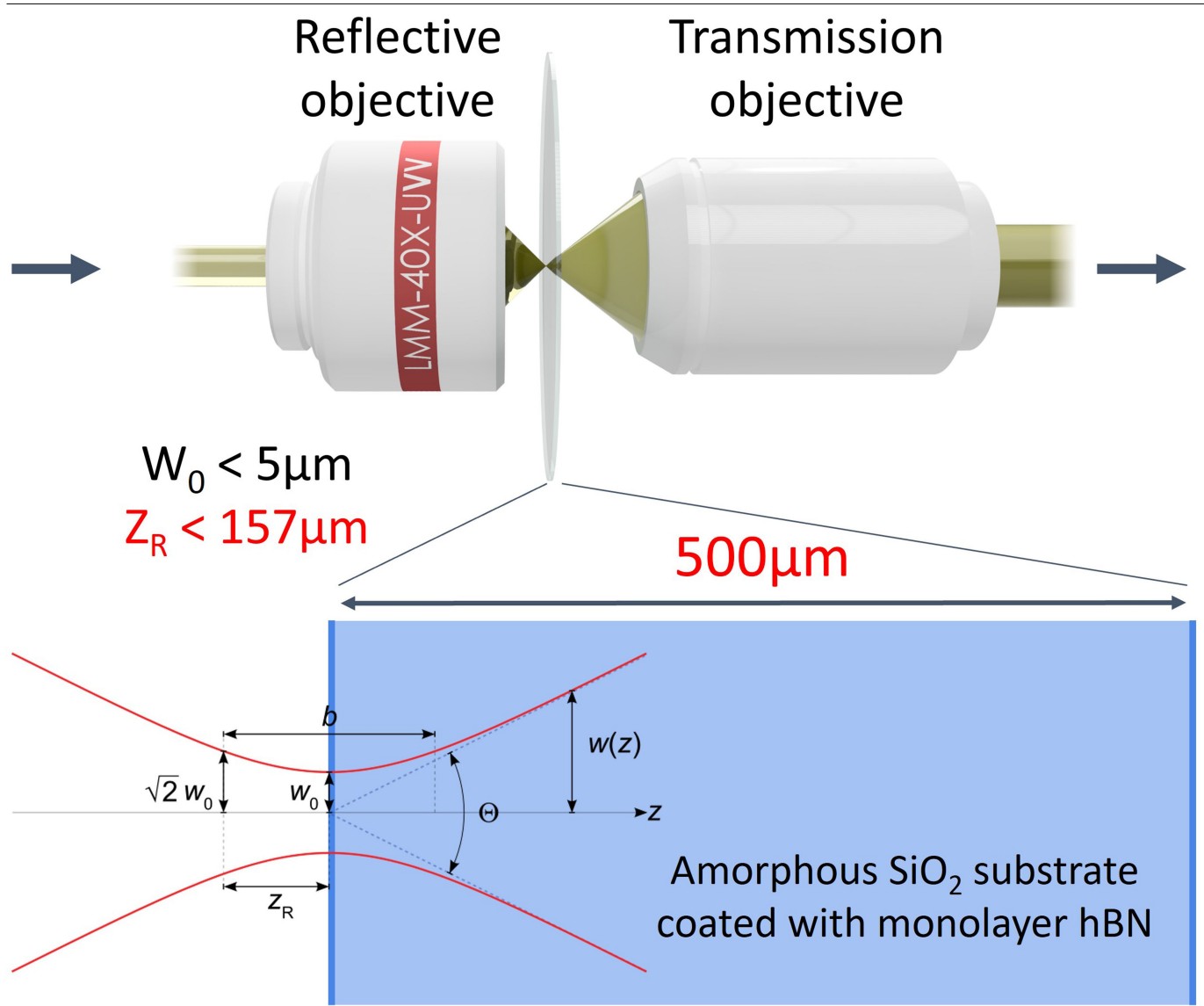

**Reflective objective**

**Transmission objective**

LMM-40X-UW

$W_0 < 5\mu m$

$Z_R < 157\mu m$

$500\mu m$

$b$

$w(z)$

$\sqrt{2}\,w_0$

$w_0$

$\Theta$

$z$

$z_R$

**Amorphous $SiO_2$ substrate coated with monolayer hBN**

**Extended Data Fig. 9 | Schematic representation of the microscope arrangement.** A reflective objective with NA: 0.4 was used to expose isolated hBN monolayer grains, coated on the front (left) surface of 500 µm thick fusJed silica substrate, with tailored laser pulses. On the output side a transmission objective of NA: 0.45 was used. This focusing configuration results in a focal spot size ($w_o$) of about 10.0 µm and Rayleigh length ($Z_R$) below 157 µm for 2 µm wavelength pump component. For 1 µm wavelength pump component these numbers are 8.0 µm and 201 µm, respectively. See Extended Data Fig. 9 for the experimental data.

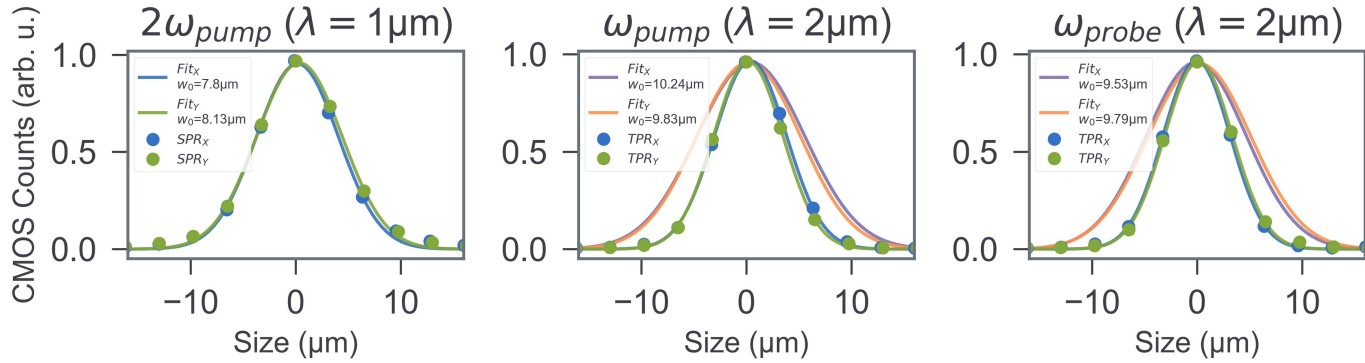

**Extended Data Fig. 10 | Beam profile cross-section.** The beam profiles of three interferometer arms were measured at the focal plane of the objective on a CMOS detector. The wavelength of $2\omega$ arm results in a single-photon response (SPR), whereas the other two arms produce a two-photon response on the detector given the band gap of silicon. To determine the waist ($\omega_0$), this was considered producing the orange and purple curves for the 2 μm cases.

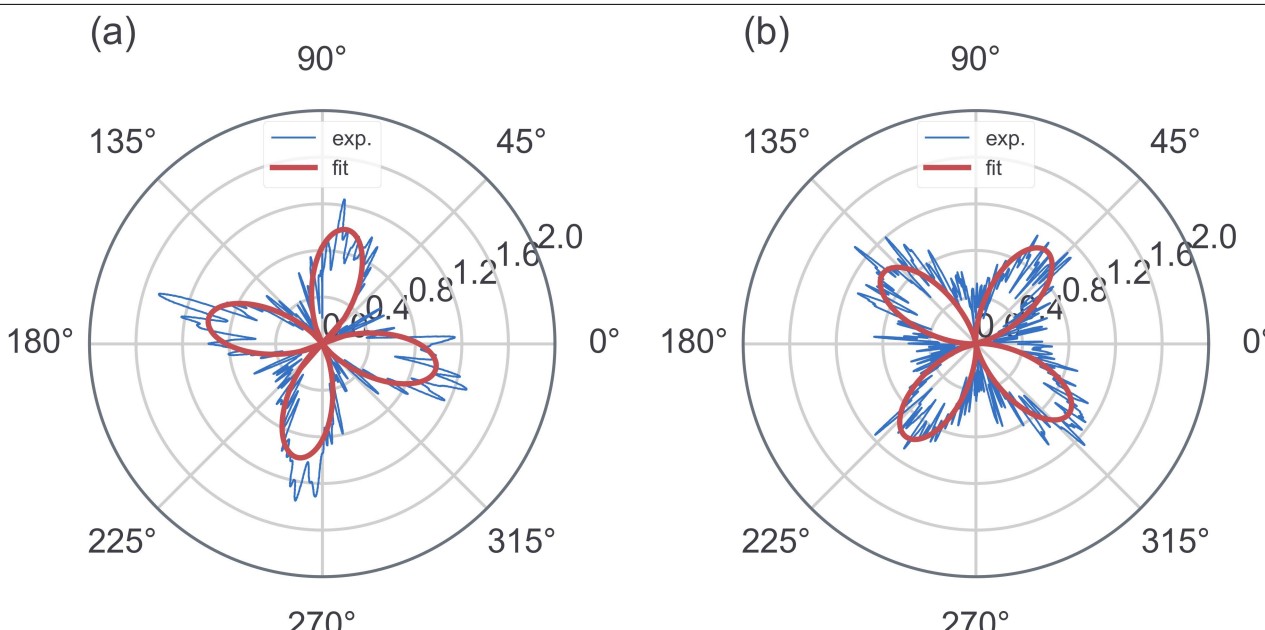

**Extended Data Fig. 11 | Polarization resolved SHG from hBN.** The rotation of SHG as a function of the polarization rotation of a linearly polarized pump pulse is illustrated here. The intensity of the SHG (normalized) is mapped after passing through a polarizer/Wollaston prism for few-odd-layer hBN patch site 1 in subplot **a** and patch site 2 in subplot **b** on a fused silica substrate. The fit function used is $Y_{fit} = A \cos(-2\theta + \varphi)^2$, where A is the amplitude of the oscillation and $\theta$ is the angle of the incoming pump polarization.

**Extended Data Table 1 | List of signals picked up by the photo-detector**

| ω | 2ω | 3ω | 4ω | 5ω |
|---|---|---|---|---|
| $Pump_\omega$ | SHG $Pump_\omega$ ($\chi^2$) | THG[†] $Pump_\omega$ ($\chi^3$) | $Pump_\omega$ ($\chi^4$) | $Pump_\omega$ ($\chi^5$) |
| $Probe_\omega$ | SHG $Probe_\omega$ ($\chi^2$) | THG[†] $Probe_\omega$ ($\chi^3$) | $Probe_\omega$ ($\chi^4$) | $Probe_\omega$ ($\chi^5$) |
| | $Pump_{2\omega}$ | Valley-Hall ω, ω, 2ω | SFG $Probe_\omega$ $Pump_{\omega, 2\omega}$ ($\chi^3$) | SFG Pump ω, ω, ω, 2ω ($\chi^4$) |
| | Leak | SFG[‡] Pump ω, 2ω ($\chi^2$) | Leak | SFG Pump $Probe_{\omega, \omega, \omega, 2\omega}$ ($\chi^4$) |
| | | SFG[*] $Probe_\omega$ $Pump_{2\omega}$ ($\chi^2$) | | SFG Pump ω, 2ω, 2ω ($\chi^3$) |
| | | Leak[†] | | SFG Probe ω, 2ω, 2ω ($\chi^3$) |
| | | FWM[**] ($\chi^3$) 2ω, 2ω, -ω | | |

These signals are sorted by their respective wavelength and non-linear processes involved in giving rise to it. Here, ω refers to λ=2 μm. The signals at frequencies other than 3ω are all rejected by high extinction ratio spectral filters. Other components are marked with a corresponding blocking method, and they are as follows: [†] Rejected by lock-in filtering or by helicity filtering of light. [‡] Rejected based on selection rules of bicircularly polarized light. [*] Rejected based on helicity filtering of light. [**] Rejected based on modulation with trefoil pump rotation. Only the signal arising from the valley hall effect is let through. For ω arm chopping, the 3ω signal possessing the helicity of 2ω pump component is detected background free. For 2ω arm chopping, the 3ω signal possessing the helicity of ω pump component is detected background free.