## [Peer Review File · Nature]

Manuscript Title: Lightwave-controlled Haldane model in monolayer hexagonal boron nitride

Reviewer Comments & Author Rebuttals

Reviewer Reports on the Initial Version:

Referees' comments:

Referee #1 (Remarks to the Author):

In the manuscript, Mitra et al. reported the optical control of time-reversal symmetry breaking and the realization of the topological model of Haldane in monolayer BN using intense trefoil structure optical field. The major conceptual advance of this work is experimental demonstration of twisting this electronic waveform of monolayer materials by tailoring the spatial symmetry of the light waveform, and subsequently, ultrafast optical control of topological properties can be realized by taking advantages ultrafast control of light. Such optical control of materials' properties on a sub-cycle timescale provides a new tool to study material physics and pave the way for potential next-generation high speed information processing device. Thus, I think the major results and claims of this manuscript will have broad interests in the scientific community and will spark intense studies along this direct in the future, which match the high standard of Nature. However, before I can make a recommendation, I think the authors should provide more experimental verification, as listed below, to support their major claims, the control of valley polarization. I also have some concerns regarding experimental details and analysis, which have to be carefully clarified. Below are detail comments for the the authors should address.

1. Is it possible to perform a comparison measurement on a valley degenerated materials that can preserve the symmetry under the excitation of tri-foil beam?

2. Why the time delay between the pump and probe was fixed at 100fs. As predicted in ref. Nature Photonics 14,728(2020), the time delay will have large influence on the ellipticity of the generated 3ω signal. The authors should provide results at different delay time, these are especially critical when the authors want to claim ultrafast control speed.

3. The electric field magnitude of the bicircular light will also have large influence on the elliptical of the generated 3ω signal. The authors should also provide pump power dependent results. It would be interesting to see the experimental verification critical signature of theoretical predictions in Figure 4 of Nature Photonics 14,728, 2020, such as Hall conductivity switches as excitation magnitude increases.

4. How the specific bicircular field, which can be well controlled by the relative intensity and phase of ω and 2ω beam,

5. The data taken from the balanced detector is questionable. To get the ellipticity of the generated 3ω , both the power of s polarized and p polarized light are needed. While the saturation of one detector makes this analysis hard to get. The authors should fix this problem. On the other hand, I suggest the authors to take the difference signal of s- and p-polarized light as the typical E-O sampling scheme of THz, which is not feasible considering one detector is saturated.

6. The authors should make sure there is no 3ω signal before the reflective objective. Also, the spectrum properties of the generated 3ω signal should be provided with time resolved characterization of the pulse (FROG for example).

7. It's not quite clear to me why 3ω signal is elliptical polarized, the authors should explain the mechanism.

There are also some other technical issues should be addressed too:

1. To transform the elliptical polarized 3ω signal to linear polarized light, the optical axis of quarter waveplate should be parallel with the long axis of the elliptical polarized 3ω signal. How to determine the correct direction of optical axis of quarter waveplate before the ellipticity is measured.

2. The authors used reflective objective to focus the bicircular pump light. However, the reflective objective has 3 arms to hold the central mirror, which will block the light in 3 directions (as illustrated in the figures below). When rotating the trefoil

bicircular pump light by tune the phase relation of ω and 2ω light, the block of

light in 3 fixed direction will result in higher power block when the trefoil is aligned with the blocked direction. This may lead to artifact in the experiment. The authors should clarify how to fix this.

Referee #2 (Remarks to the Author):

Review of the manuscript „Lightwave-controlled band engineering in quantum materials“ by S. Mitra et al.

In this paper the authors demonstrate a novel scheme to access and manipulate the valley population in a 2D material via strong-field off-resonant optical excitation, which spatial symmetry matches that of the crystal lattice of the material itself. The “tailored” strong field, off-resonant waveform of ca 30 fs duration is then twisted with respect to the material under test, breaking the time-reversal symmetry and leading to the manipulation of the bandstructure of otherwise equivalent valleys, and their population asymmetry according to the topological model of Haldane. This twist-dependent valley population asymmetry is then probed by a time-delayed optical probe pulse. The probe analysis should then yield a signal depicting the relative population of the valleys depending on the twist angle. The authors propose this method as a new, laser-based alternative to the bandstructure engineering by twisted layer stacking of 2D materials, demonstrated earlier. To demonstrate this principle, the authors have used a monolayer hexagonal boron nitride (hBN), excited via a strong ($\sim 1 \text{ V/\AA}$) lightwave composed of phase-delayed ω and 2ω signals, corresponding to $2 \mu\text{m}$ and $1 \mu\text{m}$ wavelengths. This resulted in a “trefoil” waveform matching the sublattice symmetry of hBN. The probing was performed by a linearly polarized signal at ω , yielding the THG signal at 3ω as a result of nonlinear currents induced in the material, mapping the valley populations. The readout was then performed at the 3ω frequency, and the intensity of this THG was recorded as a function of the twist angle.

In my opinion, the use of a tailored pump waveform reflecting a spatial symmetry of the material under test represents a significant inventive step.

The paper is rather well written, and the main idea and results should be accessible to a common reader familiar with modern solid state physics and modern optical spectroscopy. This paper contains 4 large figures, of them only one (Fig. 4) contains the data, whereas Figs. 1-3 describe the experiment, results of theoretical modeling, and illustrate the expected results.

The statistics in presented data is appropriate, however the filtered results presented in Fig. 4c seem “over-filtered” to me, and rather reflect the effect of the very narrow bandwidth of the applied numerical filter (see Fig. 4b).

Further, I several questions and concerns regarding the experimental method presented of this paper:

- 1) My first question is about the generality of this experimental method. As a proof of principle for their method, the authors used hBN, and hence the pump waveform was chosen to be a trefoil. Generating such a trefoil signal, and controlling its twist (here, by sub-fs-precision delay between ω and 2ω in the pump) is already quite a significant undertaking. I wonder if this method can be realistically applied to materials with more complex symmetries than hexagonal? What kind of spatial pump waveforms must to be generated, and how? I believe that this point regarding the general applicability of this method to materials with more complex crystal structures should be

openly discussed in the paper, in order to convince the reader of broader applicability of this experimental method.

2) My next comment is about the probing mechanism. Here, the THG of a time-delayed linearly-polarized probe was chosen. Wouldn't an ARPES probe be more adequate in this situation, as directly providing the information about the band structure and the valley populations after the optical pump? What other alternative probe methods could be used? I believe that such a discussion regarding alternative probing mechanisms should certainly be present in this paper.

3) Here the off-resonant strong field of the order of 1 V/\AA was used to drive the sample, both distorting the bandstructure and inducing the electronic transitions via interband tunneling. The optimal choice of both the field strength and the wavelength(s) of the pump has not been discussed in the paper. I therefore have several questions in this regard: A) What would be a figure of merit (e.g. field strength * lattice constant / bandgap size, etc) for the choice of the needed field strength? B) The same applies to the chosen wavelength(s) of the pump waveform. Would e.g. "slower" THz/FIR pump fields be more advantageous for the experiment, as providing the opportunity for the time-resolved probing with sub-cycle resolution?

4) Would other strong-field effects such as e.g. coherent Bloch oscillations, higher valley population etc may have an effect on the observations, and if yes, how to exclude them?

5) Technical comments: A) 1st page, last paragraph: "complex angle ϕ [Ref. 5]" looks like "complex angle ϕ^5 ". B) Caption of Fig. 4: (D) "function of" is repeated twice.

Referee #3 (Remarks to the Author):

This manuscript explores ultrafast light-induced switching between band structure configurations. The suggested rotating wave form (tri foil) is used to introduce modulation in the 3rd high-harmonic (HH) polarization direction as function of the direction of the excitation in an hBN sample. The method is claimed to yield both universal and robust path to engineering bands.

The Referee finds the following major concerns with this manuscript, presented claims, and observations:

[Note by the Editor: referee #3 also reviewed the paper by Biegert and coworkers for us]

0) This work is essentially a serial publication using the same tri-foil technique by some of the same co-authors: the main result - modulation of harmonics as function excitation direction – is exactly the same; also the introductory figures are essentially the same in these 2 manuscripts. Both manuscripts also claim universality – both giving just one example and very limited analysis of the harmonics. It would seem appropriate that both manuscripts should be merged and extended if

universality were to be substantiated.

1) Band-engineering aspects are inferred from the modulation the 3rd harmonic as function of the direction of the tri-foil excitation. It was early on observed that bulk HHs depend on the relative angle between the excitation direction and crystal [e.g. Nat. Phys. 7, 138 (2011)]. Later work [Nat. Photon. 1, 227 (2017)] also demonstrated that multiple excitation paths within the crystal contribute to HHs in a way that different symmetries become orthogonal (as reproduced in Fig. 4A). It seems that these well-known effects are here simply branded as band engineering in this manuscript.

2) The paper presents no evidence that dynamic bands are truly modified. The strongest connection is made by comparing experimental Fig. 4A with computed Fig. 4D; the results are completely different – why does the computation not produce the anharmonic (non-sinusoidal) features of the experiments? The narrow-band filtering procedure of Fig. 4B is not properly justified – such filtering of any measured signal will trivially produce a sinusoidal signal once Fourier transformed back to time domain with a narrow enough filter (as is done in Fig. 4B). With this analysis combination, either Fig. 4A has anharmonic artifacts or Fig. 4C has filtering artifacts – which is it and why? It is also notable that the theory disagrees qualitatively with Fig. 4A and quantitatively with Fig. 4C; both seem to indicate major discrepancies in the analysis and interpretation. This analysis seems to be the most important in the paper and its full depth should be explored and explained.

3) The paper also relies on the implicit assumption that the 3rd HH is directly connected to the band structure. It is well known [refs mentioned above] that each of the HHs reflect different aspects of the crystal symmetries and HHs have not yet delivered a direct band structure construction. To make it convincing, this work should go much beyond Ref. [PRL 115, 193603 (2015)]. Considering how weak the experiment-theory connection in the manuscript is (only the 3rd HH is analyzed, and incompletely, see comment 2), it seems a major stretch to claim that this work demonstrates universal band engineering. If this were true, the Referee would expect to see band structures, convincingly constructed from measurement, and backed up by theory. As mentioned, the connection in Fig. 4 is weak at most and does not provide evidence that bands have been engineered any differently than in the earlier works.

The manuscript also contains technical issues:

4) The manuscript talks about demonstrating “a tailored lightwave-driven analogue to twisted layer stacking.” This analogue is not properly explained, nor is it proven.

5) The manuscript state that “complex next-nearest neighbour (CNNN) hoppings are induced in the crystal through virtual nearest-neighbour hoppings.” This point of view does not seem appropriate because in semiconductors, electrons are not localized but delocalized. Of course, one can use a complete set of Wannier basis states to represent the delocalized electrons, but that does not mean that electrons are truly hopping from site-to-site.

6) To justify band engineering aspects, the authors attribute all light-induced effects to changes in the bands. How is this justified when the HH coherences decay fast?

7) It would be useful to see the full HH spectra computed and measured. How well are the

experiment and theory matching?

8) Fig. 1 – the tri-foil pulse is not clearly presented and its connection to band is not clearly made. A non-expert will definitely benefit from a more intuitive and thorough discussion.

In summary, this work lacks novelty, and merely is an extension of prior high-harmonics studies. In particular, the main claim that the bands have been engineered has not been demonstrated by the presented results. This manuscript has also major overlap with another submission by some of the same co-authors.

Author Rebuttals to Initial Comments:

Response to referees' comments

Referee #1 (Remarks to the Author):

In the manuscript, Mitra et al. reported the optical control of time-reversal symmetry breaking and the realization of the topological model of Haldane in monolayer BN using intense trefoil structure optical field. The major conceptual advance of this work is experimental demonstration of twisting this electronic waveform of monolayer materials by tailoring the spatial symmetry of the light waveform, and subsequently, ultrafast optical control of topological properties can be realized by taking advantages ultrafast control of light. Such optical control of materials' properties on a sub-cycle timescale provides a new tool to study material physics and pave the way for potential next-generation high speed information processing device. Thus, I think the major results and claims of this manuscript will have broad interests in the scientific community and will spark intense studies along this direct in the future, which match the high standard of Nature. However, before I can make a recommendation, I think the authors should provide more experimental verification, as listed below, to support their major claims, the control of valley polarization. I also have some concerns regarding experimental details and analysis, which have to be carefully clarified. Below are detail comments for the the authors should address.

Author's reply: We thank the referee for reviewing this manuscript and clearly highlighting the fact that this pioneering work has the potential to uniquely shape the study of material physics and pave the way for next generation high speed information processing. We also appreciate the realization that this study is not only a mere demonstration of a particular material or optical physics aspect, but it would spark a series of new kinds of scientific studies across different fields in future, making it very much suitable for the standard of Nature.

We also appreciate referee's suggestion for additional experimental data that unequivocally supports our claim of valley polarization control. Sparked by this comment, we performed a new campaign of experiments which have allowed us to unequivocally support our claim of valley polarization switching, as well as gain a much deeper understanding of the process (for example, extracting the intensity dependence of the valley switching due to band modification or the time scale of the valley population asymmetry in our setup. We believe the new version of the manuscript and the SI have greatly improved with these new additions.

1. Is it possible to perform a comparison measurement on a valley degenerated materials that can preserve the symmetry under the excitation of tri-foil beam?

Author's reply: We note that hBN preserves the time-reversal symmetry, and thus the valley degeneracy, for certain configurations of the trefoil field. Namely, for the pump trefoil rotation angles of 30° , 90° , 150° in Fig. 1, one can see that for these angles, the band structure of hBN is unchanged with degenerate valleys (see Fig. 1D and F). Experimentally also this was realized as one follows the zero crossing of the asymmetry curve in Fig. 3D. At those angles the right and left helicities of the 3ω probe signal show the same strength therefore no asymmetry, signalling that the pump has excited equal K and K' valley populations.

Similarly, when the power of the trefoil pulse is low, the band structure is barely modified and, as a consequence, the valleys are equally populated and the 3ω probe signal does not modulate. To

corroborate this, we performed new intensity scaling experiments that we show in Fig. 4A of the revised manuscript and attach here for convenience in Fig.R1 (see also answer to Q3). We can see that the 3ω probe signal oscillation strength gradually increases with increasing pump power, as predicted by theory. In the inset we also show how the band gap is changing after initiating from an almost degenerate (same band gap at K and K' valleys) condition.

Fig.R1 (Fig. 4A in main manuscript): Fourier amplitudes of the 3ω co-rotating signal as a function of pump intensity. Orange corresponds to the signal at 120° period, green to that at 60° period. Lines present the theoretical simulations. Inset: Pump intensity scaling of the bandgap at the K (purple-solid) and K' (green-dashed) valley for a trefoil orientation along $\theta = 120^\circ$.

Finally, for comparison with the dynamics in hBN, we also provide the contrasting results for amorphous fused silica. Fused silica lacks long-range order, such that the arrangement of the tetrahedra is random [R. P. Gupta, Phys. Rev. B 32 8278 (1985)]. Due to this random arrangement, we do not expect the trefoil field to create a momentum asymmetry that will lead to a helicity signal of the probe. Indeed, we didn't see any characteristic modulation of the generated 3ω probe helicity signal for amorphous fused silica, for the same trefoil configuration. This is illustrated in Fig. 3 A and B.

- Why the time delay between the pump and probe was fixed at 100fs. As predicted in ref. Nature Photonics 14,728(2020), the time delay will have large influence on the ellipticity of the generated 3ω signal. The authors should provide results at different delay time, these are especially critical when the authors want to claim ultrafast control speed.

Fig.R2 (Fig. 4B in main manuscript): Fourier amplitude of the 3ω co-rotating signal at 120° period as function of pump-probe delay. The dashed line is for the guide to eye.

Author's reply: In the manuscript, we have focused on the time delay between the two colors of the pump (i.e., trefoil orientation), which is the handle to manipulate the band structure. In the work the referee is referring to, we computed theoretically the anomalous Hall conductivity (AHC) during the pulse, which as the referee correctly points out and as we show in this work, can be obtained from the ellipticity/helicity of the generated 3ω probe signal in a pump-probe time-delay setup. This sort of measurement is extremely interesting when the AHC changes sign during the trefoil-matter interaction, which is indicative of the closing of the band gap. According to our simulations, our current experimental conditions do not reach that regime since the hBN band gap is too large (see inset of Fig. R1). Also, we note that pump-probe measurements which track the laser-induced AHC in time requires the probe field to be weak (so as to not to distort the trefoil waveform) and short, on the order of only a few cycles of the trefoil pump, to be able to temporally follow the change of its 3ω ellipticity (and thus the AHC) during the trefoil-matter interaction. In our current setup, the probe field is a replica of the fundamental field of the trefoil pump, so they have the same duration. We believe the results of this work will encourage further studies to probe such laser-induced closing of the band gap, which theoretically can be achieved using longer pump wavelengths and shorter probe pulses.

Stimulated by the referee's question, we have performed a pump-probe time delay experiment. In this case, as argued above, not to see the change of the AHC during the trefoil-matter interaction, but to extract the time scale of valley population relaxation in the strong-field regime, which has not been reported (this is the first work inducing valley polarization in the strong-field regime). When the probe has a large overlap with the pump at small delays, and it has a comparable strength, the spatial trefoil form of the pump will be modified, affecting the measurement and yielding a noisy modulation of the 3ω signal. If the time delay between the pump and the probe is larger than the population relaxation time T_1 or the valley scattering time, the pump-induced asymmetry in the valley population will vanish, and no modulated 3ω signal will be observed. The new pump-probe time delay experiments precisely show these effects. We show them in Fig. 4B of the manuscript and reproduced it also here in Fig.R2 for convenience. Before 60fs, when the pump and probe have a strong overlap, the Fourier spectrum is very noisy, and the points are not shown. After 120fs, the signal induced by strong-field valley polarization reduces to below noise level. The 60fs decay of the signal is fast compared to the typical valley lifetimes in one-photon resonant processes (~ 1 ps). However, we note that in our case, the measurement was performed at room temperature, and that the strong-field excitation populates a much larger area of the Brillouin zone, which can justify these fast timescales. That said, we also note that, while fast, an optical cycle of our field is around 6.6fs, which is what it takes to form the trefoil structure and create valley polarization. Thus, a sequence of few-cycle, strong tailored pulses, can in principle be used as switches of valley polarization, which we believe could be also an interesting future direction of our work. Overall, these new pump-probe time-delay measurements have made our paper even stronger, and we thank the referee for the stimulating suggestion.

Added in main text: 'In order to study the time scale of the decay of the strong-field-induced were performed at room temperature.'

3. The electric field magnitude of the bicircular light will also have large influence on the elliptical of the generated 3ω signal. The authors should also provide pump power dependent results. It would be interesting to see the experimental verification critical signature of theoretical predictions in Figure 4 of Nature Photonics 14,728, 2020, such as Hall conductivity switches as excitation magnitude increases.

Author's reply: We thank the referee again for stimulating us to do additional measurements. We now provide pump power dependence data in Fig. 4A (Fig.R1 of the reply) and an additional discussion in the main text (see also answer to Q1). As mentioned in the reply to Q2, the experimental conditions do not allow for a switching of the Hall conductivity (which will signal a band gap closing), but we do see a clear increasing modulation of the 120° and 60° oscillation amplitude of the 3ω probe signal as the trefoil pump strength is increased. Theory and experiment are in excellent agreement. This supports our claim that the band gap decreases as the pump intensity is increased. We also provide extensive theoretical simulation results in Extended Data Fig. 4 demonstrating the situations for different field intensities.

Added in main text: 'In Fig. 4A we show the scaling of the oscillation amplitude when the band gap is reduced and approaches a value comparable to the photon energy.'

Added in section 2.2.1 in Methods: 'The data shown in the manuscript ratio between the components intact.'

4. How the specific bicircular field, which can be well controlled by the relative intensity and phase of ω and 2ω beam,

Author's reply: The question is not fully understood as it is not complete. We would be very happy to answer the question in further communication if the referee still has some doubt on some aspect which is not being covered in this this reply.

5. The data taken from the balanced detector is questionable. To get the ellipticity of the generated 3ω , both the power of s polarized and p polarized light are needed. While the saturation of one detector makes this analysis hard to get. The authors should fix this problem. On the other hand, I suggest the authors to take the difference signal of s- and p-polarized light as the typical E-O sampling scheme of THz, which is not feasible considering one detector is saturated.

Author's reply: We thank the referee for stimulating us to push for the measurement of both the s- and p- polarized components. Now we have managed to do that. We completely agree that this improves the quality of the work dramatically and unambiguously demonstrates the effect we claim.

We have been able to detect the final s- and p-polarized components in a background free situation by utilizing lock-in detection while chopping both ω and 2ω arms of the interferometer independently for a single polarization configuration of the pump helicity. Earlier we were chopping the 2ω arm only. For ω arm chopping, the unwanted 3ω background having a helicity same as of ω falls on one diode (D1), and the other diode (D2) remains free from any background. Whereas for 2ω arm chopping, unwanted 3ω background having a helicity same as of 2ω falls on the D2 diode, and D1 diode remains free from any background. From these two measurements combined, keeping all the other experimental parameters unchanged, we retrieve the complete information required (see Fig. 3A and B). Here we also achieved to keep the two measurements in a single time axis by uninterrupted operation of the CW reference interferometer.

As the referee suggested, we eventually could take the difference of these detected s- and p-polarized signals and retrieve the population asymmetry at the K and K' valleys. The asymmetry, shown in Fig. 3D and attached here for convenience in Fig.R3, clearly depicts its evolution from positive to negative

values as function of trefoil rotation, signifying periodic band modification and switching between valleys.

All these new results are added in the main manuscript along with new discussion. To clearly describe the experimental procedure we have also added a complete section in the Methods.

Also, since we can now extract all the information needed to prove band modification/valley switching from an experiment with a fixed pump helicity, we have decided to remove the previous results where we showed the opposite helicity of the pump field. We believe that with these changes the manuscript has improved significantly in clarity.

Fig.R3 (Fig. 3D in main manuscript): Comparison of the asymmetries (α) deduced from the experimental and the theoretical data. The experimental curve is obtained after Fourier filtering around the frequency corresponding to the 120° oscillation, as shown in the inset.

Added in main text: ‘Experimentally, we determined both the helicity and ellipticity by is evidence of the trefoil-field-induced modification of the band structure.’

Added a new section in 2.5. in Methods: ‘However, 3ω light can be produced by between these two measurements.’

6. The authors should make sure there is no 3ω signal before the reflective objective. Also, the spectrum properties of the generated 3ω signal should be provided with time resolved characterization of the pulse (FROG for example).

Author’s reply: In the experiment it was very carefully ensured that no 3ω signal, which is produced before the hBN interaction, is detected that could contaminate the analysis. We used specialized chirp mirrors and dielectric mirrors in the ω - 2ω interferometer to get rid of the unwanted light.

We note that the 3ω signal is a symmetry-forbidden harmonic line of the ω - 2ω pump trefoil field. The possibility of its generation is minimized below noise level by optimizing the quality of circularity of the pump components. The detection of unwanted 3ω signal generated from the substrate or hBN by only pump ω or only probe ω or through two-photon process by pump 2ω and probe ω , is prohibited by combination of lock-in and polarization filtering (please also see answer of Q5).

We show additional data for the 3ω spectrum in Fig. 2C. This clearly shows that when there is no probe, no 3ω light is detected, i.e., the trefoil pulse interaction alone produces no unwanted 3ω signal.

In Table R1 we provide a table summarizing the origin of the unwanted signals and how these are prohibited to be detected. We also add this table to the Extended Data (Extended Data Table 1).

ω	2ω	3ω	4ω	5ω
Pump $_{\omega}$	SHG Pump $_{\omega}$ (χ^2)	THG † Pump $_{\omega}$ (χ^3)	Pump $_{\omega}$ (χ^4)	Pump $_{\omega}$ (χ^5)
Probe $_{\omega}$	SHG Probe $_{\omega}$ (χ^2)	THG † Probe $_{\omega}$ (χ^3)	Probe $_{\omega}$ (χ^4)	Probe $_{\omega}$ (χ^5)
	Pump $_{2\omega}$	Valley-Hall $_{\omega, \omega, 2\omega}$	SFG Probe $_{\omega}$ Pump $_{\omega, 2\omega}$ (χ^3)	SFG Pump $_{\omega, \omega, \omega, 2\omega}$ (χ^4)
	Leak	SFG ‡ Pump $_{\omega, 2\omega}$ (χ^2)	Leak	SFG Pump (χ^4) Probe $_{\omega, \omega, \omega, 2\omega}$
		SFG* * Probe $_{\omega}$ Pump $_{2\omega}$ (χ^2)		SFG Pump $_{\omega, 2\omega, 2\omega}$ (χ^3)
		Leak †		SFG Probe $_{\omega, 2\omega, 2\omega}$ (χ^3)
		FWM ** (χ^3) $_{2\omega, 2\omega, -\omega}$		

*Table R1 (Extended Data Table 1 in Extended Data): These signals are sorted by their respective wavelength and non-linear processes involved in giving rise to it. Here, ω refers to $\lambda = 2 \mu\text{m}$. The signals at frequencies other than 3ω are all rejected by high extinction ratio spectral filters. Other components are marked with a corresponding blocking method, and they are as follows: † Rejected by lock-in filtering or by helicity filtering of light. ‡ Rejected based on selection rules of bicircularly polarized light. * Rejected based on helicity filtering of light. ** Rejected based on modulation with trefoil pump rotation. Only the signal arising from the valley hall effect is let through. For ω arm chopping, the 3ω signal possessing the helicity of 2ω pump component is detected background free. For 2ω arm chopping, the 3ω signal possessing the helicity of ω pump component is detected background free.*

Regarding the FROG characterization, we believe the time resolved information for 3ω signal will not provide any additional information as we are observing the time-integrated effect of the pump. This holds because within the pump duration the pump waveform preserves one unique symmetry. The time structure of the relevant 3ω signal will be mostly governed by the input probe pulse time structure. Nevertheless, the yield of relevant 3ω signal was also too small to perform FROG characterization.

7. It's not quite clear to me why 3ω signal is elliptical polarized, the authors should explain the mechanism.

Author's reply: We thank the referee for pointing out that this was not clear. We have modified the discussion of this mechanism in the main manuscript as follows:

Added in main text: ‘We separated a small portion of the fundamental $2\ \mu\text{m}$ (ω) beam, with linear polarization and pulse duration of about 30 fs, and delayed it $\sim 60 - 120$ fs with respect to the trefoil pump pulse. The linearly polarized probe field E_{probe} generates a drift current along the polarization axis, as well as an anomalous (orthogonal) current that is proportional to the population asymmetry at the complementary valleys,

$$j_{anomalous}(t) = \frac{e^2}{\hbar(2\pi)^2} E_{probe}(t) \sum_n \int_{BZ} d\mathbf{k} \rho_{nn}(\mathbf{k}, t) \Omega(\mathbf{k}),$$

where e is the electron charge, ρ_{nn} is the population of band n , and $\Omega(\mathbf{k})$ is the Berry curvature.

The drift current is governed by the vector potential of the field, whereas the anomalous current is proportional to the electric field. This leads to a $\frac{\pi}{2}$ phase-delay difference between these components, and thus to the emission of elliptically polarized harmonic radiation. Note that the Berry curvatures around the K and K' valleys exhibit opposite signs (see Fig. 2D), $\Omega(K) = -\Omega(K')$. In absence of valley polarization for $\rho_{nn}(K) = \rho_{nn}(K')$, the anomalous current cancels, and the non-linear optical response induced by the probe field is linearly polarized. In contrast, when there is valley polarization for $\rho_{nn}(K) \neq \rho_{nn}(K')$, the anomalous current is non-zero and its sign is opposite for K and K' valley polarization. This makes the optical response of the probe elliptical, and with opposing helicity for K and K' valley polarization (see Fig. 2E-G). Hence, the valley polarization induced by the trefoil pump field is mapped onto the asymmetry $\alpha = \frac{S_1 - S_2}{S_1 + S_2}$ of the two helicity components S_1 and S_2 of the probe harmonic (3ω in present case).’

There are also some other technical issues should be addressed too:

1. To transform the elliptical polarized 3ω signal to linear polarized light, the optical axis of quarter waveplate should be parallel with the long axis of the elliptical polarized 3ω signal. How to determine the correct direction of optical axis of quarter waveplate before the ellipticity is measured.

Author’s reply: In the experiment the orientation of the Wollaston prism was kept such that the outgoing beams lie on the horizontal (‘p’ polarization) plane. With respect to that the optical axis of the $\lambda/4$ waveplate, which was placed before the Wollaston prism and after the transmission objective, was kept at 45° , i.e. 45° with respect to both ‘s’ and ‘p’ polarized direction.

In that case if the 3ω light is circularly polarized, which was the case for unwanted 3ω generation through THG of ω -pump (relevant for ω arm chopping) or pump 2ω -probe ω SFG process (relevant for 2ω arm chopping), it goes entirely on a single diode depending on the helicity. This is because the $\lambda/4$ waveplate makes it entirely either ‘s’ or ‘p’ polarized light and the Wollaston prism separates it out to ‘s’ and ‘p’ components. Therefore, the other diode remains to be free (up to noise level) from any background signal.

For the elliptically polarized 3ω light expected from the valley dynamics, after the $\lambda/4$ waveplate, a component falls on the background free diode. As we rotate the trefoil waveform keeping the probe polarization direction constant, a modulation was observed at the background free diode. This is because of the modulation of the anomalous component, which lies perpendicular to the probe polarization direction.

Now we have added a new section 1.4.2 in Methods explaining the full mechanism of optical harmonic polarimetry.

Added in Methods: '1.4.2. Probe of the valley polarization via optical polarimetry'

The authors used reflective objective to focus the bicircular pump light. However, the reflective objective has 3 arms to hold the central mirror, which will block the light in 3 directions (as illustrated in the figures below). When rotating the trefoil bicircular pump light by tune the phase relation of ω and 2ω light, the block of light in 3 fixed direction will result in higher power block when the trefoil is aligned with the blocked direction. This may lead to artifact in the experiment. The authors should clarify how to fix this.

Author's reply: In reference to the diagram provided by the referee, here the trefoil symmetry is associated with the electric field of the light, and the electric field is defined at each point on the spatial distribution of the beam. Therefore, the obstruction provided by the spider vanes does not affect the polarization symmetry of the electric field. However, as this is strong field driven process, the asymmetry in intensity might affect the result. In that context, for the reported experiments, we have used LMM25XF-UVV (Thorlabs) objective for the microscope. This objective is specially designed to minimize the effect of the spider vanes. Instead of three straight spider vanes, three curved spider vanes are used for holding the central mirror. This significantly improves any possible angular distortion at the focus. Please see the comparison diffraction plots (in log scale) in Fig. R4 provided by the company for curved (left) and straight (right) spider vanes.

Fig.R4: Comparison of diffraction plots (in log scale) for curved (left) and straight (right) spider vanes.

Source: https://www.thorlabs.com/newgrouppage9.cfm?objectgroup_id=6933

Referee #2 (Remarks to the Author):

Review of the manuscript „Lightwave-controlled band engineering in quantum materials“ by S. Mitra et al.

In this paper the authors demonstrate a novel scheme to access and manipulate the valley population in a 2D material via strong-field off-resonant optical excitation, which spatial symmetry matches that of the crystal lattice of the material itself. The “tailored” strong field, off-resonant waveform of ca 30 fs duration is then twisted with respect to the material under test, breaking the time-reversal symmetry and leading to the manipulation of the bandstructure of otherwise equivalent valleys, and their population asymmetry according to the topological model of Haldane. This twist-dependent valley population asymmetry is then probed by a time-delayed optical probe pulse. The probe analysis should then yield a signal depicting the relative population of the valleys depending on the twist angle. The authors propose this method as a new, laser-based alternative to the bandstructure engineering by twisted layer stacking of 2D materials, demonstrated earlier. To demonstrate this principle, the authors have used a monolayer hexagonal boron nitride (hBN), excited via a strong ($\sim 1 \text{ V/\AA}$) lightwave composed of phase-delayed ω and 2ω signals, corresponding to $2 \mu\text{m}$ and $1 \mu\text{m}$ wavelengths. This resulted in a “trefoil” waveform matching the sublattice symmetry of hBN. The probing was performed by a linearly polarized signal at ω , yielding the THG signal at 3ω as a result of nonlinear currents induced in the material, mapping the valley populations. The readout was then performed at the 3ω frequency, and the intensity of this THG was recorded as a function of the twist angle.

In my opinion, the use of a tailored pump waveform reflecting a spatial symmetry of the material under test represents a significant inventive step.

The paper is rather well written, and the main idea and results should be accessible to a common reader familiar with modern solid state physics and modern optical spectroscopy. This paper contains 4 large figures, of them only one (Fig. 4) contains the data, whereas Figs. 1-3 describe the experiment, results of theoretical modeling, and illustrate the expected results.

Author’s reply: We thank the referee for reviewing this manuscript and clearly identifying the ‘significant inventive step’, which can be easily accessible to the common readers in diverse scientific fields. We also appreciate referee’s comment about the clarity of writing which would be accessible to wide range of readers.

We completely agree with the referee that we did not give enough space to fully show the results. Following the referee's observation, and thanks to new measurements that we have performed, we have re-structured the manuscript. Now Fig. 1 contains the key physical idea and numerical results. Fig. 2 contains the methodology of experiment. Fig. 3 and Fig. 4 contain experimental data, including new crucial measurements of the helicity asymmetry, intensity scaling, and pump-probe delay, as well as the excellent comparison to theory.

The statistics in presented data is appropriate, however the filtered results presented in Fig. 4c seem “over-filtered” to me, and rather reflect the effect of the very narrow bandwidth of the applied numerical filter (see Fig. 4b).

Author's reply: We thank the referee for pointing this out. In the revised manuscript, the filtered i-FFT is applied only on the asymmetry parameter plotting (Fig. 3D). This is because it is the asymmetry of the 120° oscillation component that probes the band-modified-induced valley polarization, while the weaker 60° component (second harmonic Fourier component) appears due to helicity-dependent valley circular dichroism and other strong-field effects. This is now justified in the new text and in Methods section 1.4.2. In response to referee's suggestion, we take a wide Fourier filter of the dominant 120° -degree periodic peak in the FFT (see Fig. 3D inset). The result shows the expected behaviour without loss of generality. In Fig. R5, we explicitly show that if we consider smaller width of the FFT peak, still the main conclusions, i.e. periodicity of asymmetry and its values of extension ranging from positive to negative, remain intact. The last row is used in main manuscript (Fig. 3D).

In the other panels of Fig.3 in the main manuscript, we show raw experimental data with no filter or fitting, following the referee's suggestion.

Fig.R5: Inverse FFT (right) with variable (increasing from top to bottom) filter width of 120° peak in FFT spectrum (left) of asymmetry. The last row is used in main manuscript (Fig. 4D).

Further, I several questions and concerns regarding the experimental method presented of this paper:

1. My first question is about the generality of this experimental method. As a proof of principle for their method, the authors used hBN, and hence the pump waveform was chosen to be a trefoil. Generating such a trefoil signal, and controlling its twist (here, by sub-fs-precision delay between ω and 2ω in the pump) is already quite a significant undertaking. I wonder if this method can be realistically applied to materials with more complex symmetries than hexagonal? What kind of spatial pump waveforms must to be generated, and how? I believe that this point regarding the general applicability of this method to materials with more complex crystal structures should be openly discussed in the paper, in order to convince the reader of broader applicability of this experimental method.

Author's reply: We appreciate referee's concern regarding this aspect, which is very important in broader context.

While we understand the referee's query for applicability in materials with other symmetries, we want to stress that the demonstration of this effect on hexagonal materials is already extremely broad. Indeed, the vast majority of research on 2D materials and van der Waals heterostructures is based on hexagonal structures, and has opened several prolific directions on their own (e.g., twistrionics, valleytronics). Further, our scheme can in principle be applied also to change the properties of Kagome lattices [J-Xin Yin, Nature 612 647-657 (2022)].

That said, we are not limited to 2D materials. Our scheme can in principle be applied, for example, on prototypical 3D (Quantum Spin Hall) topological insulators such as Bi_2Se_3 , which shows hexagonal symmetry on every quintuple layer (see, e.g., H. Zhang et al., Nature Physics 5 438-442 (2009)). Could the breaking of time-reversal symmetry by the trefoil field lead to a controlled Quantum Spin Hall to Quantum Anomalous Hall insulating transition?

Moreover, controlled modification of the band structure with our scheme is not limited to second-neighbour hoppings. One can in principle also induce a zero first neighbour hopping, which will lead to flat bands. Admittedly, this will provide different physics than those found in the flat bands of twisted bilayer graphene [Y. Cao et al., Nature 556 43-50 (2018)], but we believe it is yet another interesting direction worth exploring.

Looking to an even broader applicability of our scheme, one can think of patterning the light such that the trefoil shape rotates spatially along the beam profile. This can be achieved, for example, using a spatial light modulator that changes the spatial phase delay between the two colors of the pump field. In this way, one can create spatially-separated regions in the material (micron-sized) with different electronic properties, in analogy to Moiré patterning.

A sentence has been added at the end of the conclusion in main manuscript to reflect this point.

Added in main manuscript: 'Indeed, we believe an interesting route is in analogy to Moiré patterning.'

2. My next comment is about the probing mechanism. Here, the THG of a time-delayed linearly-polarized probe was chosen. Wouldn't an ARPES probe be more adequate in this situation, as directly providing the information about the band structure and the valley populations after the optical pump? What other alternative probe methods could be used? I believe that such a discussion regarding alternative probing mechanisms should certainly be present in this paper.

Author's reply: As the referee pointed out, we completely agree that the probing step can be achieved in different ways. We have chosen to look at the valley polarization and measure the changes in electronic dynamics through the harmonic polarimetry because of two main reasons: (i) so that the setup is fully optical, (ii) so that we do not require a resonant pulse, as with e.g., time-resolved Faraday rotation [S. dal Conte et al., Phys. Rev. B 92 235424 (2015)], and would thus be suited for insulators such as hBN (5.9eV band gap).

While ARPES is indeed an incredibly powerful technique to observe the band structure of metals and semiconductors, it is not well suited for insulators. Furthermore, combination of ultrafast, strong,

multi-color laser fields with ARPES is extremely difficult. Our probing method, on the other hand, is valid for both semiconductors and insulators and much simpler in its implementation. We believe, however, that, while challenging, observing directly the band modification with ARPES will be an extremely interesting route to explore.

Other techniques which can be applicable to measure valley polarization are resonant X-ray absorption spectroscopy and transport valley Hall measurement. We believe that the current demonstration will vividly encourage studies with other probing mechanisms.

A sentence has been added at the discussion in main manuscript to reflect this point.

Added in main manuscript: 'All-optical probing offers simplicity, applied for probing the valley population in the present scheme.'

3. Here the off-resonant strong field of the order of 1 V/Å was used to drive the sample, both distorting the bandstructure and inducing the electronic transitions via interband tunneling. The optimal choice of both the field strength and the wavelength(s) of the pump has not been discussed in the paper. I therefore have several questions in this regard: A) What would be a figure of merit (e.g. field strength * lattice constant / bandgap size, etc) for the choice of the needed field strength? B) The same applies to the chosen wavelength(s) of the pump waveform. Would e.g. "slower" THz/FIR pump fields be more advantageous for the experiment, as providing the opportunity for the time-resolved probing with sub-cycle resolution?

Author's reply: (A) We thank the referee for stimulating us to do additional measurements, which we believe have made the paper even stronger. We provide additional experimental data demonstrating the pump intensity (field strength) scaling of the effect in Fig. 4A (see Fig. R1 in this reply) along with an additional discussion in main text. We show in the inset the theoretical prediction of how the band gap asymmetry between the valleys increases as a function increasing pump intensity, which translates into a higher valley polarization. As predicted by theory, we observe a clear increasing modulation of the Fourier amplitude of the 3ω probe signal as the trefoil pump strength is increased. Theory and experiment are in excellent agreement, further supporting our claim that the band gap decreases as the pump intensity is increased. We also provide extensive theoretical simulation results in Extended Data Fig. 4 demonstrating the situations for different field intensities. As the referee correctly points out, the lattice constant and equilibrium band gap size will also influence the band modification and valley polarization, as the derivation in Methods section 1.3 shows.

Added in main text: 'In Fig. 4A we show the scaling of the oscillation amplitude when the band gap is reduced and approaches a value comparable to the photon energy.'

Added in section 2.2.1 in Methods: 'The data shown in the manuscript ratio between the components intact.'

(B) We completely agree with the referee that longer wavelength drivers will be beneficial for such strong field band engineering. Although the experimental demonstration of it is out of the scope of the current work, the theory paper (Nat. Photon., (2020), 728-732, 14(12)) that predicted the effect by this collaboration demonstrated it for longer wavelength, and could reach the closing of the band gap with similar field intensities as the ones we use here. We also appreciate referee's view on future

experiments. Indeed, the sub-cycle resolved probing of such effect would be one of the major achievements in contemporary strong field and condensed matter physics. We believe the currently demonstrated work would be instrumental to encourage for such an experiment.

4. Would other strong-field effects such as e.g. coherent Bloch oscillations, higher valley population etc may have an effect on the observations, and if yes, how to exclude them?

Author's reply: We thank the referee for raising this important point, which we address below.

On the one hand, regarding the probing step, we note that the observation method does not rely on strong field effects, since it only requires observation of the third harmonic order. In fact, the response is fully characterized by semiclassical dynamics, where no interband current is necessary. We have now provided in the revised manuscript a clearer explanation of the measurement process:

Added in main text: 'We separated a small portion of the fundamental $2 \mu\text{m}$ (ω) beam, with linear polarization and pulse duration of about 30 fs, and delayed it $\sim 60 - 120$ fs with respect to the trefoil pump pulse. The linearly polarized probe field E_{probe} generates a drift current along the polarization axis, as well as an anomalous (orthogonal) current that is proportional to the population asymmetry at the complementary valleys,

$$j_{\text{anomalous}}(t) = \frac{e^2}{\hbar(2\pi)^2} E_{\text{probe}}(t) \sum_n \int_{\text{BZ}} d\mathbf{k} \rho_{nn}(\mathbf{k}, t) \Omega(\mathbf{k}),$$

where e is the electron charge, ρ_{nn} is the population of band n , and $\Omega(\mathbf{k})$ is the Berry curvature.

The drift current is governed by the vector potential of the field, whereas the anomalous current is proportional to the electric field. This leads to a $\frac{\pi}{2}$ phase-delay difference between these components, and thus to the emission of elliptically polarized harmonic radiation. Note that the Berry curvatures around the K and K' valleys exhibit opposite signs (see Fig. 2D), $\Omega(K) = -\Omega(K')$. In absence of valley polarization for $\rho_{nn}(K) = \rho_{nn}(K')$, the anomalous current cancels, and the non-linear optical response induced by the probe field is linearly polarized. In contrast, when there is valley polarization for $\rho_{nn}(K) \neq \rho_{nn}(K')$, the anomalous current is non-zero and its sign is opposite for K and K' valley polarization. This makes the optical response of the probe elliptical, and with opposing helicity for K and K' valley polarization (see Fig. 2E-G). Hence, the valley polarization induced by the trefoil pump field is mapped onto the asymmetry $\alpha = \frac{S_1 - S_2}{S_1 + S_2}$ of the two helicity components S_1 and S_2 of the probe harmonic (3ω in present case).'

In the theory, we can extract the k -resolved populations after the pump field (see Fig.11-L), and for a wide range of probe and pump intensity regimes we observe a one-to-one correspondence between the helicity of H3 of the probe and the valley asymmetry: the drift current is similar for all pump orientations, and the anomalous current oscillates from zero when there is no valley polarization to maximum when there is valley polarization (with π -phase change for opposite valley polarization). This relation can also be derived analytically (see Method Section, Eqs. 16-20 in Á. Jiménez-Galán, Nat. Photonics 14 728 (2020)).

On the other hand, regarding the pump mechanism, we note that valley polarization can happen via four mechanisms. Which of these mechanisms will dominate depends on the laser intensity and

frequency regime. Importantly, each of these mechanisms will show a different behaviour of our observable, highlighting its advantage.

First, in the one- (or few-) photon resonant regime, valley polarization can happen through optical valley selection rules [D. Xiao et al., PRB]. If this were the only mechanism, the helicity of H3 should not switch as a function of the pump rotation, and should not modulate or modulate only slightly. In contrast, we observe a large modulation and a switch in the helicity asymmetry as a function of the pump rotation (new Fig.3D), which discards this mechanism as the main one. As mentioned in the manuscript, this helicity-dependent mechanism has an effect in our observable: it shows as a second harmonic in the Fourier analysis of the 3ω oscillation (see Fig.1I-L and Fig.3B,C).

Second, valley polarization can happen due to the effect of the topological resonance [S. Azar Oliaei Motlagh et al., PRB 100 115431 (2019)]. This has only been studied for few-cycle circular pump fields and thus the dependence of valley polarization with a two-color counter-rotating trefoil field orientation is not known. Importantly, here the valley polarization decreases with increasing pump power (see Fig.6 of S. Azar Oliaei Motlagh et al., PRB 100 115431 (2019)). Thus, the helicity of the 3ω probe signal should also decrease for increasing pump power. We see the opposite in the experiment and theory (see Fig. 4A).

Third, due to streaking [A. Jimenez-Galan et al., Optica 8 277-280 (2021)], initially proposed for a few-cycle linearly-polarized pump field. Here, the electron populations after the pump field are streaked to the crystal momenta $k(t) = k_0 - A(t)$, where k_0 are the minimum band gap points, K and K' , and $A(t)$ is the vector potential at the maximum of the electric field. In this case, one should have populations not only at the valleys, but also at Γ (see Fig.2 of A. Jimenez-Galan et al., Optica 8 277-280 (2021)). Clearly this is not the case (see Fig.1I-L of the main manuscript). Importantly, the multi-cycle field that we are considering in this case will lead to interferences that will destroy the valley polarization. Also, the streaking condition will actually give opposite valley polarizations than those observed in Fig. 1I-L.

Fourth, valley polarization can happen due to band structure modification by the strong field [Á. Jiménez-Galán et al., Nat. Phot]. Here, (i) the helicity of the probe 3ω should oscillate strongly with 120° periodicity as a function of pump rotation and switch helicity asymmetry, as seen in the experiment, (ii) valley polarization should increase as a function of pump intensity (up to the damage threshold or the band gap closing), as observed in the new experiments (Fig.4A). Thus, this mechanism is the only one fully compatible with our experimental observations.

5. Technical comments: A) 1st page, last paragraph: “complex angle ϕ [Ref. 5]” looks like “complex angle ϕ^5 ”. B) Caption of Fig. 4: (D) “function of” is repeated twice.

Author’s reply: We thank the referee for such a careful look at the manuscript. Now these are corrected as needed.

Referee #3 (Remarks to the Author):

This manuscript explores ultrafast light-induced switching between band structure configurations. The suggested rotating wave form (tri foil) is used to introduce modulation in the 3rd high-harmonic (HH) polarization direction as function of the direction of the excitation in an hBN sample. The method is claimed to yield both universal and robust path to engineering bands.

Author's reply: We thank the referee for reviewing the manuscript and showing some of the concerns which have helped us to strengthen the paper.

The Referee finds the following major concerns with this manuscript, presented claims, and observations:

[Note by the Editor: referee #3 also reviewed the paper by Biegert and coworkers for us] This work is essentially a serial publication using the same tri-foil technique by some of the same co-authors: the main result - modulation of harmonics as function excitation direction – is exactly the same; also the introductory figures are essentially the same in these 2 manuscripts. Both manuscripts also claim universality – both giving just one example and very limited analysis of the harmonics. It would seem appropriate that both manuscripts should be merged and extended if universality were to be substantiated.

Author's reply: We do not agree with the referee's opinion of merging the two works. The two are completely independent experimental efforts (please remember these are primarily experimental papers) and differ in several crucial aspects. First, the samples are fundamentally different, one is bulk MoS₂ (an inversion-symmetric semiconductor), ours is monolayer hBN (a broken inversion symmetric insulator). Our material hBN is the prototypical pristine example of gapped (massive) graphene and therefore provides the closest condensed-matter realization possible of the tight-binding model originally proposed by Haldane, allowing for a direct and clean comparison with theory. Second, we unambiguously detect the valley polarization switching induced by the pump field by measuring the switch of the helicities of the non-linear, third order response of a probe field. This is radically different from bulk-MoS₂, which lacks Berry curvature, and therefore lacks an anomalous Hall current that can lead to such effect. As explained clearer in the revised version, in this work we are directly accessing the change in direction of the anomalous Hall current as a function of pump rotation (mapped on the switch of the 3ω helicity), as one would with, e.g., time-resolved Faraday rotation or valley Hall transport measurements, with the advantage that our scheme works in the non-resonant regime. Third, we provide the first demonstration of valleytronics and band structure engineering on an insulating 2D material, opening the way for ultrafast electronic devices consisting of combinations of 2D materials with different optical band gaps. Fourth, our new measurements show how the oscillation amplitude of the 3ω signal increases with increasing pump intensity, which is fully in accordance with theory and further confirms valley polarization induced by band structure engineering. Fifth, we provide new pump-probe time-delay scans with about 10 fs time step that allow us to estimate the timescale of relaxation of strong-field-induced valley population for the first time.

Thus, we believe that these works are largely independent and deserve consideration on their own merits. Furthermore, we strongly believe that the fact that two works by two independent experimental groups confirm a new physical mechanism (which has potential to shape future material science and strong field optical physics research, as recognized by the other referees) in two different

systems, under different physical conditions, and with two different probing mechanisms, can only be beneficial for scientific research, especially for major discoveries.

1. Band-engineering aspects are inferred from the modulation the 3rd harmonic as function of the direction of the tri-foil excitation. It was early on observed that bulk HHs depend on the relative angle between the excitation direction and crystal [e.g. Nat. Phys. 7, 138 (2011)]. Later work [Nat. Photon. 1, 227 (2017)] also demonstrated that multiple excitation paths within the crystal contribute to HHs in a way that different symmetries become orthogonal (as reproduced in Fig. 4A). It seems that these well-known effects are here simply branded as band engineering in this manuscript.

Author's reply: We feel the referee has missed the main point of our paper, and we apologize for not having made it completely clear in the previous version. The main focus and the message of this study is not high harmonic generation (HHG) spectroscopy to which the referee is referring. We fully agree that many references have demonstrated the modulation of HHs as function of crystallographic orientations, highlighting their respective crystallographic symmetries, and that there are prior studies available on HHG with trefoil driver or in general two-color drivers (in many materials). Yet, our study focuses and claims something completely different, and does not focus at all on the HHG process. We are not trying to generate or detect any harmonics of the trefoil driver (indeed, the trefoil-driver does not show a 3ω signal due to symmetry selection rules, see Fig. 2C of the revised manuscript). We are also not studying how high harmonics from the linearly polarized probe change as a function of its orientation, since the probe field remains at fixed orientation.

We feel the referee has confused our mechanism for probing valley polarization switching with a HHG study. In fact, our probing mechanism is more similar to a time-resolved Faraday rotation (TR-FR) experiment [S. dal Conte et al., Phys. Rev. B 92 235424 (2015)], but with a crucial difference. TR-FR uses a resonant probe pulse and measures the polarization of its linear response. In our case, the UV band gap of hBN forbids the use of resonant pulses, so we use a non-resonant, non-linear adaptation of TR-FR in which we measure the third harmonic response of a strong, non-resonant probe field. The non-linear dynamics leading to the third harmonic of the probe can be explained purely semiclassically (no interband dynamics necessary) as coming from a drift (intraband) current and an anomalous current (see answer to Q3).

The logic of our work is as follows: valley-contrasting band structure modification through rotation of the pump field naturally generates valley polarization, which is what we measure through the 3ω probe signal helicity. Yet, valley polarization can happen via four different mechanisms, band structure modification being just one of them. Which of these mechanisms will dominate depends on the laser intensity and frequency regime. Crucially, each of these mechanisms will show a different behaviour of our observable (3ω helicity oscillation of the probe), highlighting its advantage.

First, in the one- (or few-) photon resonant regime, valley polarization can happen through optical valley selection rules [D. Xiao et al., PRB]. If this were the only mechanism, the helicity of H3 should not switch as a function of the pump rotation, and should not modulate or modulate only slightly. In contrast, we observe a large modulation and a switch in the helicity asymmetry as a function of the pump rotation (new Fig.3D), which discards this mechanism as the main one. As mentioned in the

manuscript, this helicity-dependent mechanism has an effect in our observable: it shows as a second harmonic in the Fourier analysis of the 3ω oscillation (see Fig.1I-L and Fig.3B,C).

Second, valley polarization can happen due to the effect of the topological resonance [S. Azar Oliaei Motlagh et al., PRB 100 115431 (2019)]. This has only been studied for few-cycle circular pump fields and thus the dependence of valley polarization with a two-color counter-rotating trefoil field orientation is not known. Importantly, here the valley polarization decreases with increasing pump power (see Fig.6 of S. Azar Oliaei Motlagh et al., PRB 100 115431 (2019)). Thus, the helicity of the 3ω probe signal should also decrease for increasing pump power. We see the opposite in the experiment and theory (see Fig. 4A).

Third, due to streaking [A. Jimenez-Galan et al., Optica 8 277-280 (2021)], initially proposed for a few-cycle linearly-polarized pump field. Here, the electron populations after the pump field are streaked to the crystal momenta $k(t) = k_0 - A(t)$, where k_0 are the minimum band gap points, K and K' , and $A(t)$ is the vector potential at the maximum of the electric field. In this case, one should have populations not only at the valleys, but also at Γ (see Fig.2 of A. Jimenez-Galan et al., Optica 8 277-280 (2021)). Clearly this is not the case (see Fig.1I-L of the main manuscript). Importantly, the multi-cycle field that we are considering in this case will lead to interferences that will destroy the valley polarization. Also, the streaking condition will actually give opposite valley polarizations than those observed in Fig. 1I-L.

Fourth, valley polarization can happen due to band structure modification by the strong field [Á. Jiménez-Galán et al., Nat. Phot]. Here, (i) the helicity of the probe 3ω should oscillate strongly with 120° periodicity as a function of pump rotation and switch helicity asymmetry, as seen in the experiment, (ii) valley polarization should increase as a function of pump intensity (up to the damage threshold or the band gap closing), as observed in the new experiments (Fig.4A). Thus, this mechanism is the only one fully compatible with our experimental observations.

We apologize if the link between the measured signal and valley polarization (and, by extension, valley-selective band structure modification) was not clear in the previous version. We have now made this connection clearer in the revised text (see paragraph starting with "We separated a small portion [...]").

To finalize, the main achievement of this work is to demonstrate controllable modification of the laser-dressed band structure via spatial rotation of the polarization-shaped pulse (achieved on sub-cycle timescales), thereby allowing manipulation of the valley asymmetry and valley polarization switching. We enumerate the novelties of this work:

1. First demonstration of selective band structure modification with strong, tailored light fields, controllable on sub-laser-cycle timescales.
- 2 First realization of the model of Haldane in a pristine material with light fields, with the model parameters controlled by the parameters of the laser field.
3. First demonstration of strong-field-induced valley polarization. All previous works relied on resonant, one-photon processes to induce valley polarization with light.

4. First demonstration of valley polarization in an insulator. Since our method is non-resonant, we open the field of valleytronics to materials previously inaccessible through one-photon resonant processes, e.g., wide-gap materials such as hBN.

5. We provide the first measurement of strong-field-induced valley population relaxation.

6. We introduce a new detection technique of valley polarization, which extends time-resolved Faraday rotation to the strong-field, non-resonant regime and thus applicable to wide-gap systems.

As we hope is now clear, there is a radical difference in the motivations/aims and claims of the studies mentioned by the referee and ours. We have greatly modified the main text in the revised version, as well as performed further experiments, that have made the focus and results of this work much clearer.

2. The paper presents no evidence that dynamic bands are truly modified. The strongest connection is made by comparing experimental Fig. 4A with computed Fig. 4D; the results are completely different – why does the computation not produce the anharmonic (non-sinusoidal) features of the experiments? The narrow-band filtering procedure of Fig. 4B is not properly justified – such filtering of any measured signal will trivially produce a sinusoidal signal once Fourier transformed back to time domain with a narrow enough filter (as is done in Fig. 4B). With this analysis combination, either Fig. 4A has anharmonic artifacts or Fig. 4C has filtering artifacts – which is it and why? It is also notable that the theory disagrees qualitatively with Fig. 4A and quantitatively with Fig. 4C; both seem to indicate major discrepancies in the analysis and interpretation. This analysis seems to be the most important in the paper and its full depth should be explored and explained.

Author's reply: We thank the referee for pointing this out. The referee points out that there are strong discrepancies between experiment and theory referring to an anharmonic feature. We believe the referee is referring to the second harmonic signal of the Fourier analysis (non-harmonic signals are below noise level). We note that theory does reproduce the second harmonic signal, and due to its high relevance, we give a full analysis below and in the revised manuscript.

Before doing this, however, we want to make clear that the main observation reported (120° oscillation of the helicity-resolved harmonic signal of the *probe* as a function of the pump orientation), is not at all a trivial one, and it is a hallmark of the selective modification of the band structure (see our next answer). We stress that our experiment should not be confused with traditional HHG spectroscopy: we are not measuring the total harmonic signal, and we are not changing the orientation of the pulse that generates the harmonics, e.g., as in the works mentioned by the referee. For the modulation shown in Fig.3A to occur, the pump field must have induced a valley population asymmetry in the K or K' valleys that maximizes or vanishes with its rotation. Ab initio numerical theory and analytical theory unequivocally show that this is a consequence of selective band structure modification by the strong field. The valley polarization asymmetry shown in Fig. 3D clearly depicts its variation from positive to negative value with rotation of the trefoil, which fully establish the effect of selective asymmetric band engineering. Furthermore, the new experiments have now shown that the observed helicity modulation, which maps the valley polarization, depends dramatically on the intensity of the pump (see Fig. 4A), as also predicted by both numerical and analytical theory, which further supports our claim.

We now explain in detail the origin of the second Fourier harmonic signal to which the referee refers to. The new simulation results provided in Fig. 3C (also see Extended Data Fig. 4) clearly depicts the contribution of the second harmonic signal (60° periodicity). This is further confirmed by the discrete Fourier transform shown in the inset, which can be directly compared with its experimental counterpart shown in Fig. 3B. The reason behind the second Fourier component (60° oscillation) can be seen by looking at Fig.1 (earlier Fig.2 panels F-I), panels I-L. These four panels show the valley polarization for trefoil oriented along to 30°, 60°, 90° and 120° in Fig.3C, respectively. Panels J and L are separated by 60° but have clearly not exactly opposite valley polarizations. The main reason is the influence of the pump helicity, as mentioned in the manuscript. This difference makes the curve in Fig. 3C not a perfect sinusoid. Moreover, this second harmonic signal depends strongly on the intensity of the trefoil pump pulse, as we now show in Fig. 4A (also see Extended Data Fig. 4). As the pump intensity becomes stronger, so does the second harmonic signal relative to the first, as observed in the experiment and excellently reproduced by the theory. Following the referee's suggestion, we have added all these details regarding the observed second harmonic signal in the main manuscript and we have made a dedicated section in Methods section 1.4.2.

Added in 1.4.2. in Methods:

1.4.2. Probe of the valley polarization via optical polarimetry

To transfer these valley populations into an optical degree of freedom that can be measured in the experiment, we use a linearly-polarized probe pulse after the bicircular pulse which, as explained in the main text, allows to map them into the helicity of its non-linear harmonic response (H3 in this case). As in the experiment, we use a probe light field of the same ω frequency as the fundamental field in the bicircular pulse. The field strength of the probe field is $F_{\text{probe}} = 0.01$ a.u, it is polarized along the Γ -M direction (x-direction in the figures), and its duration is 30 fs of full width at half maximum. We tested different probe field strengths, but it did not affect our results.

Extended Data Fig. 4 shows the results of the polarimetry analysis. The two helical components that the two photo-diodes measure are defined as,

$$\begin{aligned}\hat{e}_l &= \frac{1}{\sqrt{2}}(\hat{e}_x + i\hat{e}_y), \\ \hat{e}_r &= \frac{1}{\sqrt{2}}(\hat{e}_x - i\hat{e}_y).\end{aligned}\tag{22}$$

Therefore,

$$\begin{aligned}\hat{e}_l &= \left| \frac{A_x}{\sqrt{2}} e^{i\phi_x} + \frac{A_y}{\sqrt{2}} e^{i(\phi_y + \pi/2)} \right|^2, \\ \hat{e}_r &= \left| \frac{A_x}{\sqrt{2}} e^{i\phi_x} + \frac{A_y}{\sqrt{2}} e^{i(\phi_y - \pi/2)} \right|^2,\end{aligned}\tag{23}$$

where A_x, A_y (ϕ_x, ϕ_y) are the Fourier amplitudes (phases) of the harmonic of interest of the probe (H3 in this case), along direction x and y, respectively. We can re-write the above as

$$\begin{aligned}\hat{e}_l &= \frac{1}{2}|A_x|^2 + \frac{1}{2}|A_y|^2 + |A_x||A_y| \cos(\phi_x - \phi_y - \pi/2), \\ \hat{e}_r &= \frac{1}{2}|A_x|^2 + \frac{1}{2}|A_y|^2 + |A_x||A_y| \cos(\phi_x - \phi_y + \pi/2),\end{aligned}\tag{24}$$

The phase ϕ_y changes by π as the valley polarization changes sign since the y-component of the current comes from the anomalous contribution and thus is proportional to the Berry curvature, which has the same magnitude but opposite sign in both valleys. As the valley polarization changes upon rotation of the bicircular field, the interference (cosine) term in the expression above oscillates sinusoidally, switching sign with a switch in valley polarization. Additionally, the interference term of the two helicities oscillates out of phase because of the factor $\pm\pi/2$. Extended Data Fig. 4 shows the interference term of the two helicity components, which indeed oscillate sinusoidally and out-of-phase. As expected, these interference terms are maximum or minimum when there is maximum valley polarization and zero when there is none. The asymmetry of these two interference terms completely characterizes the valley polarization.

Yet, the signal observed in the experiment also includes the amplitude term in the equation above, i.e., $\frac{1}{2}|A_x| + \frac{1}{2}|A_y|$, which also oscillates. Its oscillation comes from unequal population injection as a function of rotation, and other non-linear effects occurring during the harmonic generation. However, this term is the same for both helicities and thus it is merely a background introducing higher order Fourier components in the oscillation. We plot this term in Extended Data Fig. 4.

The helicity signal is then the sum of the amplitude term, which is common to both helicities, and the interference term, which is different for each helicity and which contains the information on the valley polarization. The total left and right signal gives the red and green curves, respectively, shown in Extended Data Fig. 4 and also in the main text, which is the curve to be compared with the experiment.

It is important to note that regardless of the value of the amplitude term, since it is a background common to both helicities, we can remove its influence. For this, we Fourier filter the oscillation in order to extract only the 120° periodic oscillation. In this way, the asymmetry of the two helicity signals after Fourier filtering characterizes the valley polarization, with the change of sign indicating valley switching.

Added in main text: 'When the laser irradiates only the fused silica the band gap is reduced and approaches a value comparable to the photon energy.'

3. The paper also relies on the implicit assumption that the 3rd HH is directly connected to the band structure. It is well known [refs mentioned above] that each of the HHs reflect different aspects of the crystal symmetries and HHs have not yet delivered a direct band structure construction. To make it convincing, this work should go much beyond Ref. [PRL 115, 193603 (2015)]. Considering how weak the experiment-theory connection in the manuscript is (only the 3rd HH is analyzed, and incompletely, see comment 2), it seems a major stretch to claim that this work demonstrates universal band engineering. If this were true, the Referee would expect to see band structures, convincingly constructed from measurement, and backed up by theory. As mentioned, the connection in Fig. 4 is weak at most and does not provide evidence that bands have been engineered any differently than in the earlier works.

Author's reply: This is not accurate, and we believe stems from the same confusion we referred to in our answer to Q2. We do not rely on the assumption that the 3rd harmonic is directly connected to band structure. In fact, we do not assume this at all. The purpose of the harmonic analysis here and that in the referred PRL is quite different. In the PRL (see Fig. 1), the strong field excites the electron

from the valence to the conduction band and drives it to other crystal momenta and higher energy points on the conduction band landscape. When it recombines with the hole at the valence band, away from the lowest bandgap momentum, it emits high energy HHs. Therefore, different HHs correspond to different energy and momentum points.

In our case, we are not detecting any harmonics from the strong trefoil driver. The 3rd harmonic of the linearly polarized probe does not necessarily excite electrons from the valence to the conduction band, but rather generates a non-linear, third-order response that is well described within a semiclassical approximation (where interband processes are neglected), and which contains the information on the valley physics. We fully explain this now in the revised manuscript. We copy below the relevant paragraph:

Added in main text: ‘We separated a small portion of the fundamental 2 μm (ω) beam, with linear polarization and pulse duration of about 30 fs, and delayed it ~60 - 120 fs with respect to the trefoil pump pulse. The linearly polarized probe field E_{probe} generates a drift current along the polarization axis, as well as an anomalous (orthogonal) current that is proportional to the population asymmetry at the complementary valleys,

$$j_{\text{anomalous}}(t) = \frac{e^2}{\hbar(2\pi)^2} E_{\text{probe}}(t) \sum_n \int_{\text{BZ}} d\mathbf{k} \rho_{nn}(\mathbf{k}, t) \Omega(\mathbf{k}),$$

where e is the electron charge, ρ_{nn} is the population of band n, and Ω(k) is the Berry curvature.

The drift current is governed by the vector potential of the field, whereas the anomalous current is proportional to the electric field. This leads to a $\frac{\pi}{2}$ phase-delay difference between these components, and thus to the emission of elliptically polarized harmonic radiation. Note that the Berry curvatures around the K and K' valleys exhibit opposite signs (see Fig. 2D), $\Omega(K) = -\Omega(K')$. In absence of valley polarization for $\rho_{nn}(K) = \rho_{nn}(K')$, the anomalous current cancels, and the non-linear optical response induced by the probe field is linearly polarized. In contrast, when there is valley polarization for $\rho_{nn}(K) \neq \rho_{nn}(K')$, the anomalous current is non-zero and its sign is opposite for K and K' valley polarization. This makes the optical response of the probe elliptical, and with opposing helicity for K and K' valley polarization (see Fig. 2E-G). Hence, the valley polarization induced by the trefoil pump field is mapped onto the asymmetry $\alpha = \frac{S_1 - S_2}{S_1 + S_2}$ of the two helicity components S₁ and S₂ of the probe harmonic (3ω in present case).’

The manuscript also contains technical issues:

4. The manuscript talks about demonstrating “a tailored lightwave-driven analogue to twisted layer stacking.” This analogue is not properly explained, nor is it proven.

Author’s reply: Here these two are connected analogically. The primary purpose of twisted layer stacking is to generate new band structural properties. With stacking two or more single-layers, the new band structure emerges which is different than that of the individual layers. Moreover, the twisting of one layer relative to the other provides more control to realize different band structure.

Here we realize similar final goal with replacing one of the layers by structure light, i.e. a ‘stacking’ of one layer of material and one ‘layer’ of light. Moreover, like twistrionics, we achieve new band configurations by twisting the light structure relative to the material structure. However, the later light-

material configuration can be beneficial as it is fast, transient, and reversible, which is not the case for conventional twistrionics with two (or multiple) layers of material.

5. The manuscript state that “complex next-nearest neighbour (C>NN) hoppings are induced in the crystal through virtual nearest-neighbour hoppings.” This point of view does not seem appropriate because in semiconductors, electrons are not localized but delocalized. Of course, one can use a complete set of Wannier basis states to represent the delocalized electrons, but that does not mean that electrons are truly hopping from site-to-site.

Author’s reply: Monolayer hBN is an insulator with a bandgap of about 6 eV. The tight binding approximation has been successfully applied in insulators (and semiconductors) in numerous works. It has been used many times to describe graphene and gapped graphene, for which hBN is the prototypical pristine example [A.H. Catro Neto, Rev. Mod. Phys. 81 109 (2009)]. Haldane's celebrated model, for instance, uses such tight-binding description, and we have adopted it to show the comparison between the light-dressed system and his model.

6. To justify band engineering aspects, the authors attribute all light-induced effects to changes in the bands. How is this justified when the HH coherences decay fast?

Author’s reply: We apologize once more for not having been clear regarding the probing mechanism. The coherences of the interband processes of high harmonic generation are not relevant to our work (see our answers to previous questions). We are only sensitive to the valley population lifetime (i.e., T_1 , not T_2). This includes both the time it takes the electrons to scatter from one valley to another (inter-valley scattering) and population relaxation. We have now performed additional measurements to estimate this value, which has not been reported for strong-field-induced valley polarization. We find that the signal drops to below noise level in around 50fs after the end of the pump pulse. This is considerably faster than typical valley lifetimes with one-photon resonant pulses, but it is expected since: (i) our experimens are performed at room temperature, (ii) the strong field injects electrons in a wider region of the Brillouin zone (see, e.g., Fig.1I-L).

7. It would be useful to see the full HH spectra computed and measured. How well are the experiment and theory matching?

Author’s reply: Since this was not intended as a work of HHG spectroscopy, we do not have a full HH spectra, nor we believe it is necessary for our purposes since it does not bear connection to the physics we are trying to describe. There are already several works before that have showed HHG spectroscopy in monolayers.

Our measurement relies on the polarization of the third harmonic of the *probe* field. We have confirmed that the trefoil pump field produces the expected $3N+1$, $3N-1$ harmonic lines, with forbidden $3N$ lines. In Fig.2C of the revised manuscript we now show the signal at 3ω when only the pump is present, which is below noise level, and when both pump and probe are present, which shows a large signal. We attach it below for convenience (Fig.R6).

Fig.R6 (Fig. 2C in main manuscript): 3ω yield as function of wavelength for pump (blue) and probe (red) irradiation.

8. Fig. 1 – the tri-foil pulse is not clearly presented and its connection to band is not clearly made. A non-expert will definitely benefit from a more intuitive and through discussion.

Author's reply: We now refer to the Methods when we introduce the trefoil field. In the Methods, we give a thorough discussion of the field, including its mathematical expression. In the text, we show the connection between the trefoil and the band structure in Fig.1. Fig. 1A and B shows the electric field structure of the light pulse including its time progression. In C, the Lissajous figure of the electric field is depicted in reference to the real-space hBN lattice. In D-G we show the Lissajous figure of the vector potential of the field in reference to the reciprocal-space hBN lattice. We also present results of calculations along with the electric field and lattice configurations, and associated band modification and population inversion in Fig. 1D - L.

9. In summary, this work lacks novelty, and merely is an extension of prior high-harmonics studies. In particular, the main claim that the bands have been engineered has not been demonstrated by the presented results. This manuscript has also major overlap with another submission by some of the same co-authors.

Author's reply: We believe that the referee's opinion and final decision stems from a big misconception of our work, as we have thoroughly argued in our previous answers. Our study is far from an extension of prior high-harmonic studies; it is not a high-harmonic study. It is the first demonstration of selective band structure modification with strong, tailored light fields, controllable on sub-laser-cycle timescales. It is the first realization of the model of Haldane in a pristine material with light. It is the first demonstration of strong-field-induced valley polarization. It is the first demonstration of valley polarization in an insulator. It now provides the first measurement of strong-field-induced valley population relaxation. It introduces a new detection technique of valley polarization, which extends time-resolved Faraday rotation to the strong-field regime. Importantly, the concept of a shaped lightwaveform controlling the band structure of 2D materials depending on its rotation opens new possibilities in material engineering, where ideas from lightwave electronics merge with those from twisted heterostructures.

We have also counter-argued that the observable we have chosen is indeed an accurate indicator of band structure engineering, which is further confirmed by the new experiments we have performed. In fact, we believe that, aside from time-resolved ARPES, which is extremely challenging for this

insulating material and with this field configuration, our probing mechanism is the most reliable indicator for band engineering. The behaviour of this observable as a function of pump rotation and pump intensity allowed us to demonstrate that the only mechanism of valley polarization compatible with the experimental results, and fully supported by theory, is selective band structure modification.

Reviewer Reports on the First Revision:

Referees' comments:

Referee #1 (Remarks to the Author):

The revised version has addressed my major concerns in last round review. I only have the following minor point for the authors to address before a final recommendation to publish.

1. I suggest the authors add detail experimental characterization results of the tri-foil beam after the coherent synthesis in the supporting information.
2. My previous question 4 should be: "4. How the specific bicircular field, which can be well controlled by the relative intensity and phase of ω and 2ω beam, affects the optical control."

Referee #2 (Remarks to the Author):

In the revised version of the manuscript, the authors have generally addressed all the questions I raised in my initial review of the original submission.

One of my questions was about the generality of this method, and its broader applicability to crystal structures other than hexagonal. Even though the authors chose not to state it explicitly in their rebuttal, from their reply it follows that the demonstrated experimental method can in principle be applied ONLY to hexagonal crystals, and the demonstrated trefoil waveform is the ONLY one possible. Since this is a very important piece of information, this fundamental limitation must be honestly emphasized and explained early on in the manuscript, perhaps even in its title: "Lightwave-... in quantum materials with hexagonal structure".

My other questions: regarding overfiltering of the data, alternative probing mechanisms, quantitative effect of strong field on the electronic structure of the material, and other possible effects such as Bloch oscillations etc, have been answered in a satisfactory manner.

--- Comments on authors' rebuttal to referee #3 ---

1) Reviewer point 1. I generally agree with the claim of Reviewer regarding the limited universality of the presented method. The reply of the authors goes very deep into detail and difference between the present sample (hBN) and the one from their previous paper (MoS₂). In my opinion, the main limitation is more general: the presented method is strictly limited to the combination of a trefoil waveform with a hexagonal material, and cannot be realistically applied to a material with any other symmetry.

2) Reviewer point 2. I partly agree with the critique of the Reviewer regarding oversimplified Fourier-filtering of the data. The whole procedure could be made more transparent by the authors.

3) Reviewer point 3. I cannot make a definite statement regarding the proper use of the term "band engineering". In the recent years this term has been used in too many cases, including bandstructure modification by light not bearing any practical benefit for the optical/electronic properties of the material under test. Within this broader definition of "band engineering", the demonstrated effect can be definitely considered as such. However, if one thinks of bandstructure engineering in the sense of "deterministic designer bandstructure", this experiment falls somewhat short of it in my opinion. However it could be just a first step in this direction.

Author Rebuttals to First Revision:

Referee #1

The revised version has addressed my major concerns in last round review. I only have the following minor point for the authors to address before a final recommendation to publish.

Answer: We thank the referee for raising these major concerns. Addressing those concerns surely improved the quality of the paper.

1. I suggest the authors add detail experimental characterization results of the tri-foil beam after the coherent synthesis in the supporting information.

Answer: We thank the referee for suggesting adding experimental characterization of the trefoil beam. We provide the spectral and temporal characteristics of the beam components in Extended Data Fig. 7, focus size characteristics in Extended Data Fig. 10. Other than that, we also provide the trefoil induced third harmonic generation data in Fig. 3c in main text, whose zero yield signifies the quality of circularity of the beam component during coherent synthesis of the trefoil waveform.

2. My previous question 4 should be: "4. How the specific bicircular field, which can be well controlled by the relative intensity and phase of ω and 2ω beam, affects the optical control."

Answer: Thank you for clarifying this important question. The relative phase between the ω and 2ω beams gives the rotation of the trefoil, that we scan in the experiment. The relative intensity between the two colors in this specific (inversion-symmetry-broken) material controls the asymmetry in the valley oscillation. This is because each color has an opposite helicity, and thus carries an opposite optical absorption valley-selection rule. Even if the effect of the optical absorption valley-selection rule is small compared to the bandgap modification effect described in the manuscript, it does have an impact on the oscillation asymmetry of the valley state, i.e., on the appearance of a 60° component, as discussed in the revised version of the manuscript. Additionally, we have provided detailed simulation results on how different intensities of the trefoil beam affect the optical control in Extended Data Fig. 3 and 4, which support the pump intensity dependent observations presented in Fig. 4a.

Referee #2:

In the revised version of the manuscript, the authors have generally addressed all the questions I raised in my initial review of the original submission.

Answer: We thank the referee for pointing out important questions and issues. Addressing those concerns surely improved the quality of the paper.

One of my questions was about the generality of this method, and its broader applicability to crystal structures other than hexagonal. Even though the authors chose not to state it explicitly in their

rebuttal, from their reply it follows that the demonstrated experimental method can in principle be applied ONLY to hexagonal crystals, and the demonstrated trefoil waveform is the ONLY one possible. Since this is a very important piece of information, this fundamental limitation must be honestly emphasized and explained early on in the manuscript, perhaps even in its title: "Lightwave-... in quantum materials with hexagonal structure".

Answer: We thank the referee for stressing this point. While the current scheme is in principle not only applicable to hexagonal systems, we have not explored other types of crystal symmetry. However, we believe, for example, that the trefoil field could be applied to rhombohedral crystals that are formed by stacked hexagonal planes of atoms, such as in the 3D topological insulator Bi_2Se_3 . Exploring the effect of the trefoil field on these types of systems is clearly a very interesting avenue for the future. Additionally, one can tailor the field such that it has a $n+1$ -fold symmetry by a combination of a circular fundamental frequency and its counter-rotating n -th harmonic. However, one must take into account that in the present case, one is tailoring the inversion and time-reversal symmetries of the crystal, which lead to many of the interesting electronic and topological properties of 2D crystals. One would have to study other types of materials case by case. In this regard, we have softened the claim of generality in the abstract, introduction and conclusion. We now only refer to light-driven modification of valley-selective bandgap in hBN. We also appreciate the referee's suggestion of modifying the title of the manuscript, which is now "Lightwave-controlled Haldane model in monolayer hexagonal boron nitride". We believe that with these changes our claims throughout the manuscript are now clear and concrete.

My other questions: regarding overfiltering of the data, alternative probing mechanisms, quantitative effect of strong field on the electronic structure of the material, and other possible effects such as Bloch oscillations etc, have been answered in a satisfactory manner.

Answer: We are very glad to address the concerns satisfactory, which actually improved the quality of the paper many folds.

Comments on authors' rebuttal to referee #3:

1) Reviewer point 1. I generally agree with the claim of Reviewer regarding the limited universality of the presented method. The reply of the authors goes very deep into detail and difference between the present sample (hBN) and the one from their previous paper (MoS₂). In my opinion, the main limitation is more general: the presented method is strictly limited to the combination of a trefoil waveform with a hexagonal material, and cannot be realistically applied to a material with any other symmetry.

Answer: We thank the referee for commenting on the response provided to Referee #3. Indeed, as we mention in our reply above, we agree with the referee that, at this point, the scheme is only thought-out to be relevant to 2D hexagonal systems (monolayer and multi-layer). In these systems, the electronic and topological properties are governed by both their inversion and time-reversal symmetries, which is what we control with the field.

2) Reviewer point 2. I partly agree with the critique of the Reviewer regarding oversimplified Fourier-filtering of the data. The whole procedure could be made more transparent by the authors.

Answer: This point of overfitting was addressed carefully in the revised version of the manuscript and the previous reply. The main observation does not depend on the fitting; it is now directly retrieved from the raw data and its Fourier transform. We thank both the referees for pointing this out.

More specifically, now the filtered i-FFT is applied only on the asymmetry parameter plotting (Fig. 3D). This is because it is the asymmetry of the 120° oscillation component that probes the band-modified-induced valley polarization, while the weaker 60° component (second harmonic Fourier component) appears due to helicity-dependent valley circular dichroism and other strong-field effects. This is now justified in the text and in Methods section 1.4.2. In response to referee's suggestion, we take a wide Fourier filter of the dominant 120-degree periodic peak in the FFT (see Fig. 3D inset). The result shows the expected behaviour without loss of generality. In Fig. R1, we explicitly show that if we consider smaller width of the FFT peak, still the main conclusions, i.e. periodicity of asymmetry and its values of extension ranging from positive to negative, remain intact. The last row is used in main manuscript (Fig. 3D).

In the other panels of Fig.3 in the main manuscript, we show raw experimental data with no filter or fitting, following the referee's suggestion.

Fig.R1: Inverse FFT (right) with variable (increasing from top to bottom) filter width of 120° peak in FFT spectrum (left) of asymmetry. The last row is used in main manuscript (Fig. 4D).

- Reviewer point 3. I cannot make a definite statement regarding the proper use of the term "band engineering". In the recent years this term has been used in too many cases, including bandstructure modification by light not bearing any practical benefit for the optical/electronic properties of the material under test. Within this broader definition of "band engineering", the demonstrated effect can be definitely considered as such. However, if one thinks of bandstructure

engineering in the sense of "deterministic designer bandstructure", this experiment falls somewhat short of it in my opinion. However it could be just a first step in this direction.

Answer: We completely understand the referee's point. We use "band engineering" in the sense that we are able to tune the band gap magnitude, selectively at each valley, and its curvature, with the lightwave of the field. This ultimately stems from the control of fundamental symmetries that we are able to have with the tailored light, and the fact that in 2D materials, these fundamental symmetries control the electronic and topological properties especially at the valley band gap. We cannot with this method, for example, change a semiconducting band gap from indirect to direct. Strictly speaking, our method provides lightwave control of the valley bandgap, but in its current form does not offer the possibility to achieve a custom-made band structure. We believe, however, that our work will open the way for these types of studies in the future. Given the rightful concerns, we have softened our claims of "band engineering" in the abstract, and introduction, and we now refer to it as "valley-selective bandgap modification"

E.g. in abstract: *'The universality and robustness of our scheme opens the way to valley-selective band gap engineering on the fly, unlocking the possibility to create few-femtosecond switches of quantum degrees of freedom.'*

In introduction: *'By rotating the structured lightwave in space, we demonstrate sub-cycle-controlled TR symmetry breaking and valley band gap control in an insulating hBN monolayer, realizing the light-driven analogue to the topological model of Haldane.'*

We have also modified the title of the paper to be more specific, as suggested by the editor and the referee. Now it reads *'Lightwave-controlled Haldane model in monolayer hexagonal boron nitride'*.